# Double exchange interaction in Mn-based topological kagome ferrimagnet
**Jiameng Wang**[1,2], **Arthur Ernst** [3,4] ✉, **Victor N. Antonov**[5], **Qi Jiang** [6] ✉, **Haoji Qian**[7], **Deyang Wang**[1,2], **Jiefeng Cao**[8], **Fangyuan Zhu**[8], **Shan Qiao** [1,2,9] ✉ & **Mao Ye** [1,2,8] ✉

Recently discovered Mn-based kagome materials, such as $RMn_6Sn_6$ (R = rare-earth element), exhibit the coexistence of topological electronic states and long-range magnetic order, offering a platform for studying quantum phenomena. However, understanding the electronic and magnetic properties of these materials remains incomplete. Here, we investigate the electronic structure and magnetic properties of $GdMn_6Sn_6$ using x-ray magnetic circular dichroism, photoemission spectroscopy, and theoretical calculations. We observe localized electronic states from spin frustration in the Mn-based kagome lattice and induced magnetic moments in the nonmagnetic element Sn experimentally, which originate from the Sn-$p$ and Mn-$d$ orbital hybridization. Our calculations also reveal ferromagnetic coupling within the kagome Mn-Mn layer, driven by double exchange interaction. This work provides insights into the mechanisms of magnetic interaction and magnetic tuning in the exploration of topological quantum materials.

The interplay of magnetism and electronic states in strongly correlated systems has remained at the frontier in condensed matter physics. Transition metal-based kagome magnets offer a fertile ground for investigating various topological quantum magnetic phases[1–8]. This is primarily due to their distinctive corner-sharing triangular lattice structure and the presence of time-reversal symmetry breaking, such as $Mn_3Sn$, $Co_3Sn_2S_2$[2–7]. Over the past few decades, extensive research has focused on the kagome-type rare-earth manganese-based compound family $RMn_6Sn_6$ (R, rare-earth elements). This family exhibits a natural layered structure, strong spin-orbit coupling effect (SOC) and the coexistence of ferromagnetic order. These characteristics make them promising candidates for realizing the quantum anomalous Hall effect (QAHE)[9,10].

The magnetic properties of $RMn_6Sn_6$ are peculiar and diverse among the various R atoms, as evidenced by neutron diffraction[11,12]. Contemporary research on $RMn_6Sn_6$ compounds primarily focuses on the interplay between their complex magnetism and the topological kagome electronic bands. For heavier rare-earth elements (R = Gd-Ho), $RMn_6Sn_6$ exhibits ferrimagnetic behavior below the Curie temperature. This phenomenon is confirmed to generate massive Dirac fermions

characterized by varying Chern gaps, exemplified by the out-of-plane easy axis of magnetization in $TbMn_6Sn_6$[13]. Additionally, most $RMn_6Sn_6$ compounds exhibit temperature-dependent spin reorientation from out-of-plane to in-plane, except for $GdMn_6Sn_6$, which primarily shows in-plane magnetism. Transport studies have highlighted that the significant anomalous Hall effect and quantum oscillations in $YMn_6Sn_6$ and $GdMn_6Sn_6$, attributed to nontrivial Berry curvature and the Dirac linear dispersion band near the Fermi level ($E_F$), furthering our understanding of the interplay between macroscopic magnetism and material microstructure[10,14]. Concurrently, a theoretical model suggests that the exchange coupling interaction between the $4f$ electrons of rare-earth elements and Mn $3d$ electrons decreases as the atomic number of R increases within the $RMn_6Sn_6$ family[13,15], pointing to a potential mechanism for modulating the topological properties of these quantum magnets. However, the mechanism in the formation of long-range magnetic order in $RMn_6Sn_6$ family is still unclear.

The establishment of long-range ordering requires the itinerant magnetism of valence band electrons, and the magnetic order is closely related to the unique highly localized flat band generated by the Mn-kagome

[1]State Key Laboratory of Functional Materials for Informatics, Shanghai Institute of Microsystem and Information Technology, Chinese Academy of Sciences, Shanghai, 200050, PR China. [2]Center of Materials Science and Optoelectronics Engineering, University of Chinese Academy of Sciences, 100049 Beijing, PR China. [3]Institute for Theoretical Physics, Johannes Keppler University Linz, Altenberger Strasse 69, A-4040 Linz, Austria. [4]Max-Planck-Institut fur Mikrostrukturphysik, Weinberg 2, D-06120 Halle, Germany. [5]G.V. Kurdyumov Institute for Metal Physics of the N.A.S. of Ukraine, 36 Academician Vernadsky Boulevard, UA-03142 Kyiv, Ukraine. [6]Center for Transformative Science, ShanghaiTech University, Shanghai, 201210, PR China. [7]Hangzhou Institute of Technology and School of Microelectronics, Xidian University, Hangzhou, 310000, PR China. [8]Shanghai Synchrotron Radiation Facility, Shanghai Advanced Research Institute, Chinese Academy of Sciences, Shanghai, 201204, PR China. [9]School of Physical Science and Technology, ShanghaiTech University, Shanghai, 201210, PR China. ✉e-mail: Arthur.Ernst@jku.at; jiangq@sari.ac.cn; qiaoshan@mail.sim.ac.cn; yem@sari.ac.cn

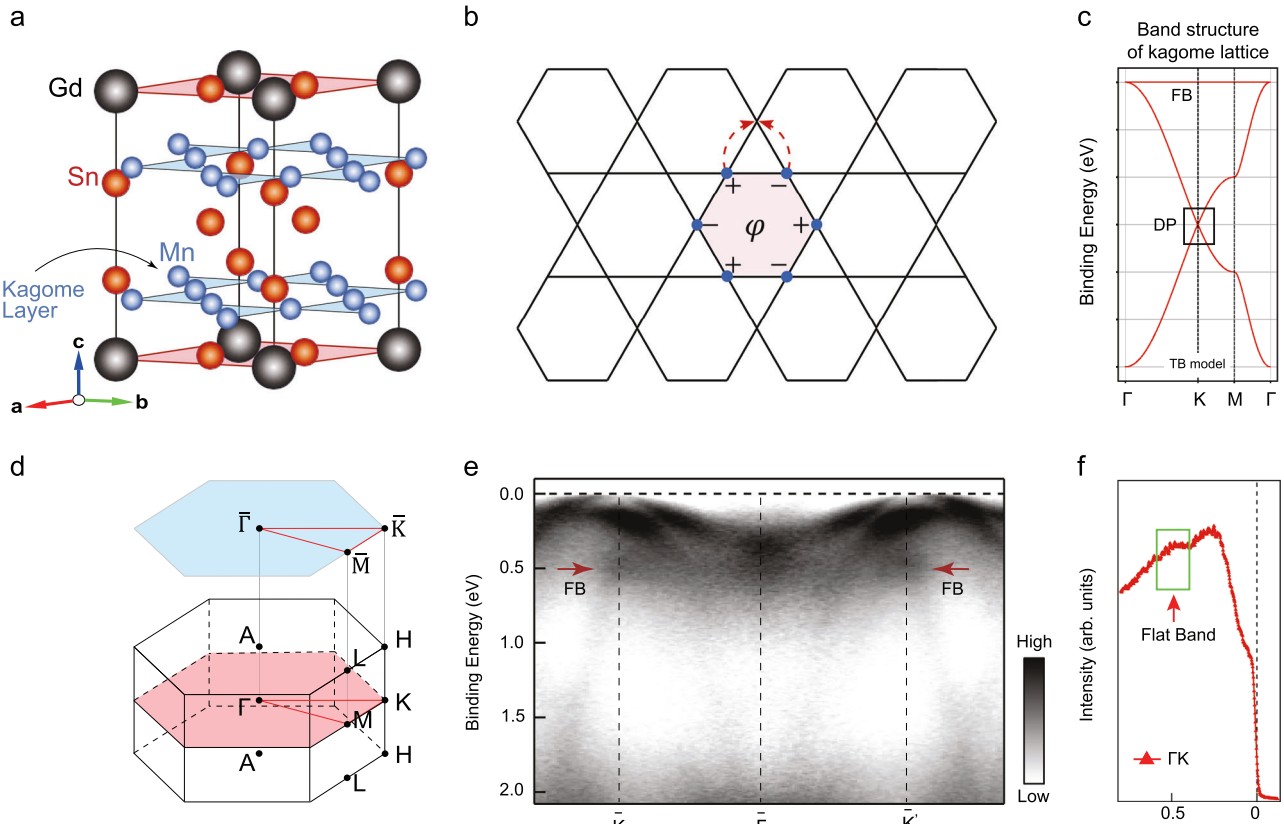

**Fig. 1 | Crystal and electronic structures of bulk single-crystal GdMn₆Sn₆.** $\mathbf{a}$ The crystal structure of GdMn₆Sn₆. Kagome lattice structure formed by Mn (blue) atoms in the a-b plane, sandwiched between Gd (black) and Sn (red) atoms. $\mathbf{b}$ The sketch illustrates the tight-binding (TB) model on a kagome lattice with nearest-neighbor (NN) in-plane hopping. $\mathbf{c}$ Schematic 2D band structure of kagome lattice from TB model. $\mathbf{d}$ Bulk and projected surface Brillouin Zone (BZ) of GdMn₆Sn₆ with marked high-symmetry points. $\mathbf{e}$ Experimental band structure along the $\bar{K} - \bar{\Gamma} - \bar{K}$ high symmetry. The flat band appears clearly, as indicated by the red arrows. $\mathbf{f}$ Energy distribution curves (EDC) from experiments along the $\bar{\Gamma} - \bar{K}$ direction, showing the flat band approximately 0.5 eV below the Fermi level ($E_F$).

geometric lattice in RMn₆Sn₆. This localization competes with the itinerant electrons in the formation of the magnetic ordered ground states in GdMn₆Sn₆, making the investigation of the electronic states particularly meaningful. Although neutron diffraction experiments have revealed ferromagnetic coupling within the kagome-Mn layer[11,12], the relationship between these electronic states and magnetism remains unclear. Furthermore, the magnetic properties such as the magnetic anisotropy, are also significantly influenced by the rare-earth elements and the coupling between Mn and rare-earth layers, indicating that the 4$f$ electrons of the rare-earth elements also participate in the magnetically ordered coupling. However, the investigation on the mechanism of magnetic properties, especially the exchange interactions between R-Mn and the magnetic coupling of RMn₆Sn₆ are still lacking from either experimental or theoretical aspects[16,17], which is critical for exploring their topological properties and the development of applications based on magnetic kagome materials.

In this work, we investigate the magnetic interaction and associated electronic structure by employing element-resolved x-ray magnetic circular dichroism (XMCD) and angle-resolved photoemission spectroscopy (ARPES), combined with density functional theory (DFT) calculations in GdMn₆Sn₆. Both ARPES and DFT calculations confirmed that the flat band characteristic of GdMn₆Sn₆ results from the localized Mn atoms within the kagome layer. Most significantly, we have detected induced magnetic moments on the Sn sites, which are anti-parallel to those on the Mn atoms and parallel to those on the Gd atoms, thus confirming the ferrimagnetic structure of GdMn₆Sn₆. Moreover, our comprehensive calculations shed light on the magnetic coupling in this material. This coupling is attributed to double exchange interaction between Mn atoms, facilitated by hybridization with Sn-$p_z$ orbitals, which is key to the development of itinerant magnetism

in GdMn₆Sn₆. Our findings provide a fresh perspective on the interaction mechanism of microscopic magnetic structures in the RMn₆Sn₆ family.

## Results
### The crystal and electronic structures of GdMn₆Sn₆
GdMn₆Sn₆ single crystal exhibits a layered hexagonal structure with the space group P6/$mmm$ (No. 191), characterized by lattice parameters ($a = b = 5.47\,\text{Å}$, $c = 8.92\,\text{Å}$). This structure is composed of manganese kagome planes, with tin (Sn) and gadolinium (Gd) distributed across different layers along the $c$-axis, as illustrated in Fig. 1a. GdMn₆Sn₆ is defined by its in-plane ferrimagnet ground state (Curie temperature, $T_c = 440\,\text{K}$), where the Gd-Mn atoms are collinear antiferromagnetically coupled. As shown in Fig. 1b, in a kagome lattice, the wave functions of adjacent sites on a hexagonal ring have opposite phases. This destructive results in a zero transition probability between nearest-neighbor (NN) sites, leading to electron localization within the hexagonal ring. The band structure of the 2D kagome lattice, considering NN hopping and calculated using the tight-binding (TB) model (see Supplementary Note 5), is depicted in Fig. 1c[18,19]. It displays a Dirac cone at the $K$ point, a saddle point at the $M$ point, and a flat band extending across the entire Brillouin Zone (BZ), indicating strong localization and correlation of electrons. The photon energies corresponding to these high-symmetry points were determined through photon energy-dependent experiments in the Supplementary Note 2. Also, we investigated the electronic structure of the Fermi surface in GdMn₆Sn₆. We present the ARPES band structure and DFT calculations in the $k_z = \pi$ plane in the Supplementary Note 3, where the Fermi surface clearly exhibits a six-fold symmetric structure, consistent with the crystal symmetry. The band structure observed experimentally along the $\bar{M} - \bar{K} - \bar{\Gamma} - \bar{K} - \bar{M}$

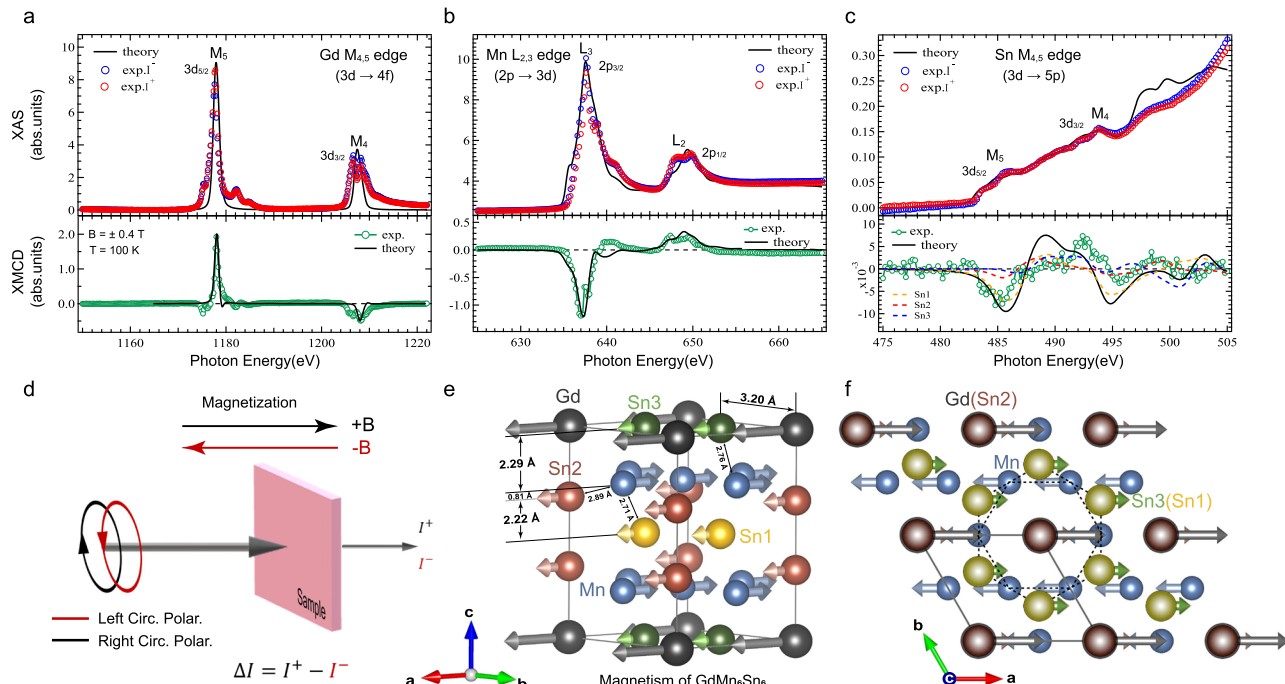

**Fig. 2 | XAS and XMCD spectra of GdMn$_6$Sn$_6$ samples at Gd M$_{4, 5}$ edges, Mn L$_{2, 3}$ edges and Sn M$_{4, 5}$ edges. a–c** Experimental and calculated x-ray absorption spectroscopy (XAS) and x-ray magnetic circular dichroism (XMCD) spectra of Gd, Mn and Sn element. I$^+$ (red circles) and I$^-$ (blue circles) represent the absorption intensities measured under positive and negative magnetic fields, respectively. The XMCD data which shown as green circles are obtained by calculating the difference between I$^+$ and I$^-$ absorption spectra. **d** Schematic of the experimental setup. The absorption intensity

of the sample under positive (+B, black) and negative (−B, red) magnetic fields corresponds to left-circular (red) and right-circular (black) polarized light, respectively. The XMCD intensity is defined as $\Delta I = I^+ - I^-$. **e, f** Schematic of the microscopic magnetic structure in GdMn$_6$Sn$_6$ showing that magnetic moments on Mn are anti-parallel coupled with those on Sn and Gd. Different atoms are labeled with different colors, with Sn1, Sn2, and Sn3 shown in yellow, red, and green, respectively. Arrows indicate the magnitude and direction of the magnetic moments of the different atoms.

direction at a photon energy of $h\nu = 94$ eV is shown in Fig. 1e. Figure 1f presents the integrated energy distribution curve (EDC) along the $\bar{\Gamma} - \bar{K}$ direction, clearly revealing the flat band located at 0.4–0.6 eV below $E_F$. This is consistent with the flat band results along the $\bar{\Gamma} - \bar{A}$ direction shown in the Supplementary Note 2c, d. Orbital-resolved band structure calculations in the Supplementary Note 4 confirm that the flat band originates from Mn 3$d$ orbital electrons and exhibits orbital hybridization with Sn 5$p$ electrons. In the GdV$_6$Sn$_6$ material related to GdMn$_6$Sn$_6$, resonant ARPES has also demonstrated the hybridization of V-3$d$ and Sn-5$p$ orbitals, which can promote the formation of long-range magnetic order[20].

**Signature of induced XMCD peaks of non-magnetic Sn atoms**

Furthermore, we investigated the formation mechanism of the long-range magnetic order of GdMn$_6$Sn$_6$ utilizing x-ray absorption spectroscopy (XAS) and x-ray magnetic circular dichroism (XMCD). These techniques are highly effective for analyzing electronic and magnetic structures due to their element-specific excitation capabilities, which enable the identification of the magnetic contributions of each constituent to the total magnetism[21,22]. XMCD experiments are based on the principle of magnetic circular dichroism. Magnetic samples exhibit different absorption intensities under different polarized light due to optical selection rules. This phenomenon occurs because polarized light breaks the system symmetry, leading to selective excitation of electronic states. Experimentally, switching the direction of the magnetic field is equivalent to reversing the direction of polarization. The magnetic field changes the electronic states through Zeeman splitting, thereby producing a difference in the absorption spectra, which is used to analyze the magnetic properties of the material. Figure 2d illustrates the experimental setup. The XAS spectra of the sample are obtained using circularly polarized light with positive (black) and negative (red) polarization or by applying an external magnetic field parallel (or antiparallel) to the direction of the light beam. The difference between these spectra constitutes the XMCD spectrum. Since the easy magnetization

direction lies in-plane, we aligned the incident x-ray beam at a 60-degrees angle relative to the sample surface normal direction. The experiments were conducted at 100 K. In addition, we performed DFT calculations of XAS and XMCD spectra within an linear muffin-tin orbital (LMTO) method[23,24] utilizing the same DFT functional as for electronic and magnetic structure calculations reported above.

Figure 2a displays the Gd 3$d \to$ 4$f$ transition in XAS and XMCD spectra for GdMn$_6$Sn$_6$. The multi-peak structures in the XAS spectra around photon energies approximately ($h\nu$) $\sim$1207.4 and 1177.6 eV are attributed to the excitations from the Gd 3$d_{5/2}$ and 3$d_{3/2}$ core levels, respectively. In Fig. 2b, the $2p^63d^5 \to 2p^53d^6$ transition at Mn L$_{2,3}$ edges of XAS and XMCD spectra is observed, corresponding to the $2p_{3/2}$ (649.2 eV) and $2p_{1/2}$ (637.6 eV) core excitations, respectively. These peak structures and positions are consistent with those in other Mn-based materials[25]. Our theoretical spectra (shown as black solid lines) match these experimental results very well, with the XMCD results at Mn L$_{2,3}$ edges being particularly close to the theoretical predictions for Mn$^{2+}$ ions. Moreover, we analyzed the XAS spectra and XMCD spectra at the non-magnetic element Sn 3$d \to$ 5$p$ absorption edges shown in Fig. 2c. Although the XAS spectra intensities at Sn M$_5$ (485.0 eV) and M$_4$ (493.2 eV) edges under different polarizations are relatively weak, clear XMCD signals were observed in the ferrimagnetic states of GdMn$_6$Sn$_6$. These induced small XMCD signals at the non-magnetic Sn-site matched the theoretical calculations quite well, further confirming the M$_{4,5}$ edges originate from its induced intrinsic magnetic moments.

Especially, for the Gd 3$d \to$ 4$f$ transition, the sign in XMCD is positive in the M$_5$ edge and negative in the M$_4$ edge. In contrast, the Mn 2$p \to$ 3$d$ transition, the sign is negative in the L$_3$ edge and positive in the L$_2$ edge. According to the selection rules, the opposite XMCD sign for these edges suggests an anti-parallel coupling between the Gd 4$f$ and Mn 3$d$ spins, experimentally proving that GdMn$_6$Sn$_6$ is a ferrimagnetic structure[26,27]. Simultaneously, the direction of XMCD signals at the Mn L$_{2,3}$ edges and Sn

**Table 1 | Magnetic moment (μB) of Mn and Sn elements at different sites in DFT calculations**

| ATOMS | S | P | D | TOTAL |
|---|---|---|---|---|
| Mn | 0.010 | 0.008 | 2.341 | 2.359 |
| Sn1 | −0.030 | 0.165 | 0.010 | −0.185 |
| Sn2 | −0.021 | −0.144 | 0.005 | −0.160 |
| Sn3 | −0.022 | −0.144 | 0.008 | −0.158 |

$M_{4,5}$ to the $d \to p$ transition and $p \to d$ transition process, respectively, which indicates the antiparallel coupling between the magnetic moments of Mn and Sn. The calculation results demonstrate three different sites in a unit cell at Sn atoms with yellow, red, and blue dotted lines as shown in Fig. 2c and Table 1. We illustrate the microscopic magnetic structure of $GdMn_6Sn_6$ based on the experimental magnetic coupling directions of different elements, as shown in Fig. 2e, f. The magnetic moments of Gd $f$ electrons and Sn $p$ electrons couple antiferromagnetically with Mn $d$ electrons. In a unit cell, the Sn1, Sn2, and Sn3 atoms are each in different chemical environments. The Sn1 site is sandwiched between the upper and lower Mn kagome layers, allowing it to directly participate in the Mn-Sn-Mn magnetic exchange interaction[28]. It is also spatially closer to the Mn atoms, resulting in the largest induced magnetic moment. Although the Sn2 site is close to the Mn kagome layer in the $c$ direction, it is pushed to the outside of the unit cell by the Gd atoms, thus reducing its induced magnetic moment. The Sn3 atoms are influenced by the Gd atoms, which weakens their induced magnetic moments. This complex magnetic structure indicates that the hybridization of valence band electrons may be fundamental to understanding the origin of the magnetic mechanism in $GdMn_6Sn_6$.

## Exchange interaction strength calculations

In order to investigate the magnetic coupling mechanism in $GdMn_6Sn_6$ we applied the magnetic force theorem as it is implemented within the multiple scattering theory[29,30]. Understanding the strength of interaction within the kagome lattice is essential for unraveling the magnetic properties of this material. Due to the weak SOC effects in $GdMn_6Sn_6$ and the isotropic exchange interaction within the Mn layers, we adopt the Heisenberg model to describe the spin correlations instead of the Kitaev model[31]. We focused on the exchange constants $J_{i,j}$ in the classical Heisenberg Hamiltonian, defined as:

$$\hat{H} = -\sum_{i,j} J_{i,j} \vec{S}_i \cdot \vec{S}_j$$

Figure 3 shows the exchange interaction strength $J$ (in meV) between intralayer and interlayer Mn-Mn, Gd-Mn atoms in the real-space supercell, with and without the Hubbard $U$ correction, respectively, where the axes (in unit of Å) indicate the relative lattice position in real space from the central atom. The color scale in red and blue represents positive (ferromagnetic coupling) and negative (antiferromagnetic coupling) values of $J$ between this site atom with the central atom, respectively. As shown in Fig. 3a, three distinct Mn sites in a primitive cell are identified in the kagome Mn-Mn layer, labeled as Mn1, Mn2, and Mn3. The interaction strength between its nearest neighbor sites is denoted as $J_{NN-12}$ and $J_{NN-13}$. The lattice exhibits $C_2$ symmetry on a very small spatial scale. The localized Mn atoms in the kagome layer exhibit antiferromagnetic coupling. The exchange interaction strength map also shows $C_2$ rotational symmetry, where $J_{NN-12} = J_{NN-13}$, consistent with the symmetry of the lattice structure. Due to the strong localization of Mn-3d electrons, it is necessary to consider the Coulomb

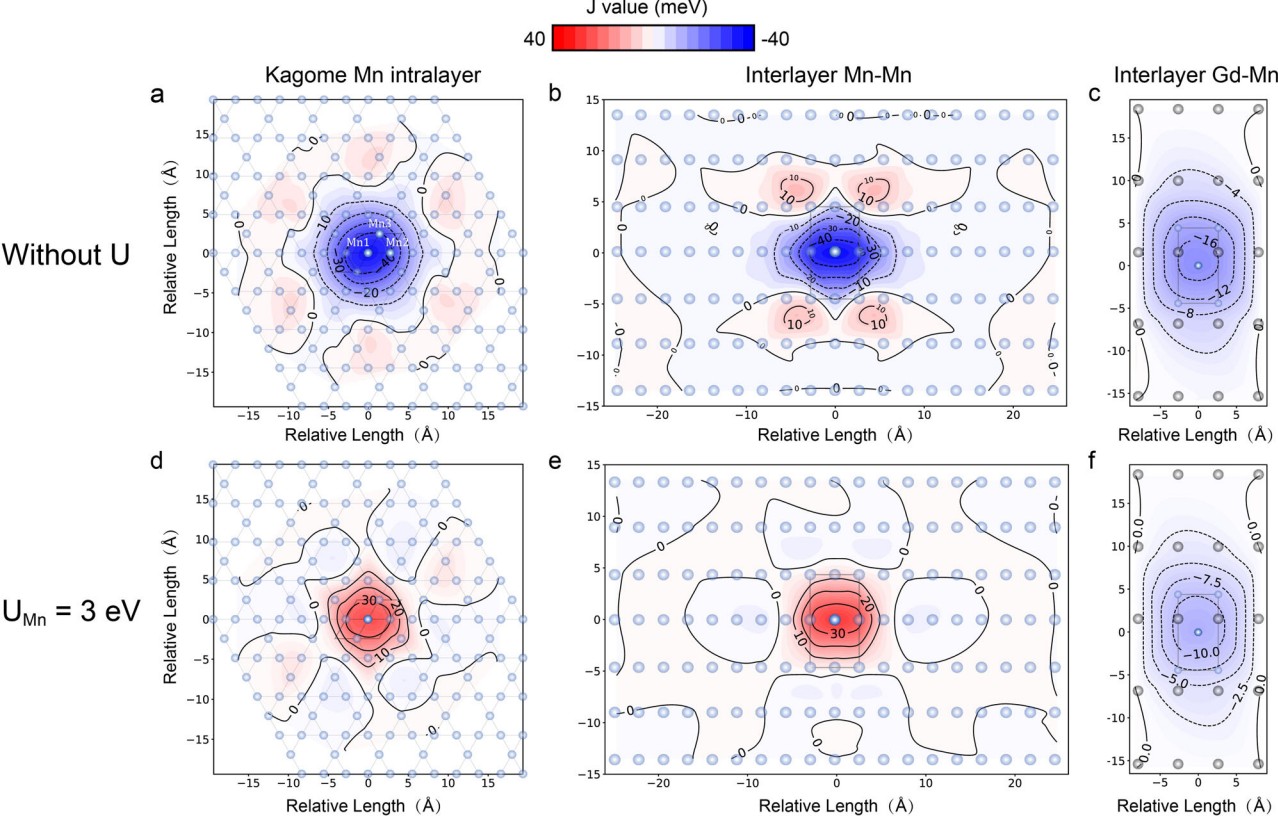

**Fig. 3 | The distribution map of magnetic exchange constants $J$ (in meV) in $GdMn_6Sn_6$. a–c** Calculated without considering the effect of Coulomb interaction $U$. **d–f** Calculated with considering the Coulomb interaction $U_{Mn}$ = 3 eV. The exchange coupling strength in intralayer kagome Mn-Mn, interlayer Mn-Mn and Gd-Mn within a supercell were calculated respectively. The color map represents the exchange coupling strength $J$, with positive (ferromagnetic) values shown in red and negative (antiferromagnetic) values in blue. Contour lines indicate the regions of constant exchange coupling strength. The Coulomb interaction $U$ significantly alters the ferromagnetic exchange interaction within the Mn-Mn kagome lattice.

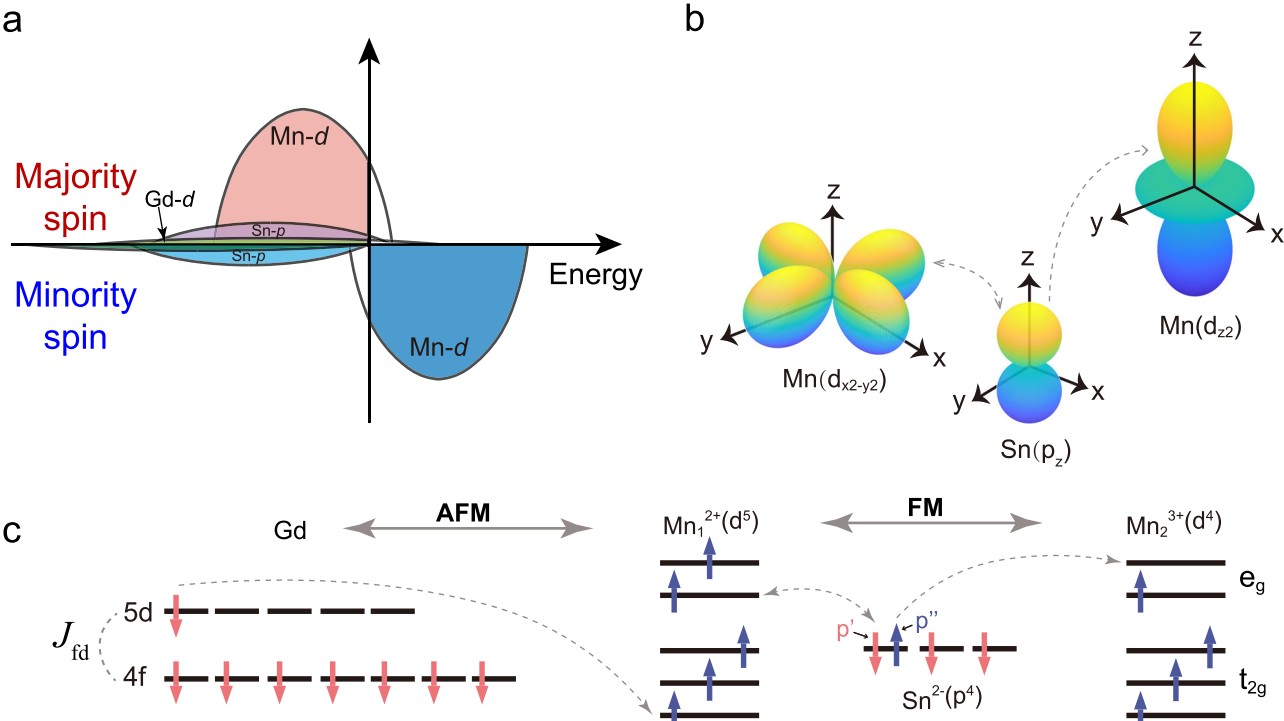

**Fig. 4 | Schematic illustrations of double exchange interaction in GdMn₆Sn₆. a** A sketch of the mechanism of magnetic interaction based on calculated density of states (DOS). The majority spin and minority spin states of the Mn-$d$, Sn-$p$ and Gd-$d$ orbitals are shown with different colors. The hybridization between Mn-$d$ and Sn-$p$ orbitals causes their spins to align antiparallel. **b** Illustration of $p$-$d$ orbital hybridization between Mn-Sn atoms. Considering symmetry, the Mn $d_{x^2-y^2}$ and $d_{z^2}$ orbitals can only interact with the Sn $p_z$ orbitals, forming the basis of magnetic interaction in GdMn₆Sn₆. **c** Depiction of the magnetic moments and exchange interaction between Gd and Mn atoms. The antiferromagnetic (AFM) interaction between Gd and Mn is mediated by the $J_{fd}$ exchange interaction, while ferromagnetic (FM) coupling between Mn atoms is facilitated by $p$-$d$ hybridization involving the Sn $p_z$ orbitals. $p'$ and $p''$ represent the virtual electron hopping processes in the double exchange interaction, respectively.

interaction $U$, as depicted in Fig. 3d. Specifically, the Coulomb interaction $U$ causes the exchange integral strength between Mn atoms to shift from negative to positive due to the charge transfer between Mn atoms, while the distribution of $J$ maintains $C_2$ symmetry. Same-spin electron transitions between different energy levels of Mn atoms, leading to ferromagnetic coupling between Mn atoms within the kagome layer and reducing the absolute value of the exchange interaction strength. This results in a slight disparity between $J_{NN-12}$ and $J_{NN-13}$. As the distance from the central Mn atom increases, the discrepancy between the second and third nearest neighbors $J_{NN-12}$ and $J_{NN-13}$ gradually diminishes until the exchange interaction of $J$ becomes negligible and can be disregarded. Compared with the interlayer Mn atoms, the kagome Mn have stronger correlation interaction due to spin frustration caused by the triangular lattice structure.

Moreover, Fig. 3b, e reveal that the exchange interaction strength between interlayer Mn-Mn atoms also shows weak asymmetry along the $c$-axis direction. This is attributed to the crystal structure of GdMn₆Sn₆ which is sequentially stacked along the $c$-axis as Mn-Gd (Sn) -Mn-Sn-Sn-Sn-Mn. Although the Mn sites are equivalent, the interlayer Mn-Mn exchange integrals exhibit asymmetry due to the separation of Mn layers by distinct Sn layers. The interaction strength rapidly decays beyond a single primitive cell period, indicating the presence of weak interlayer coupling in GdMn₆Sn₆. Therefore, the kagome Mn layer serves as an easy cleavage plane. Here, the interlayer Mn atoms are coupled in parallel, while the Gd-Mn atoms across layers show consistent antiparallel coupling, the Coulomb interaction $U$ appears to exert minimal influence on this coupling behavior. These observations are consistent with the XMCD results.

For a deeper understanding of the ferrimagnetic order formation in GdMn₆Sn₆, we analyzed the density of states (DOS) from first-principles calculations (see Supplementary Materials) in Fig. 4a, and found the hybridization of Sn $p$ states and Mn $d$ states pushes the majority spin component of Sn $p$ states above $E_F$. This shift reduces the occupation of Sn $p$ majority spin states, resulting in antiparallel magnetic moments between Sn and Mn atoms. Such hybridization, through its covalent mechanism, diminishes local magnetization and system energy, thereby leading to the antiparallel alignment observed in our XMCD experiments. Simultaneously, using the Disordered Local Moment (DLM) method to simulate the magnetic properties of GdMn₆Sn₆ in a high-temperature disordered state also verified the presence of indirect double exchange interaction. In the DLM model, thermal perturbations are introduced, assuming that the magnetic moments in the system are disordered, meaning that the magnetic moment of each atom is assumed to be random. This allows us to obtain the electronic structure and energy distribution in the disordered magnetic state. By comparing the total energy and atomic magnetic moment distribution at T = 0 K with those calculated using the DLM method, it was found that the system tends to form a magnetically ordered state at low temperatures, which is consistent with the ferromagnetic ordering induced by double exchange interaction.

Figure 4b highlights the indirect ferromagnetic exchange interaction among Mn-Sn-Mn atoms, where the Sn $p_z$ orbital is the dominant component in the band structure. The symmetry principle indicates that this $p_z$ orbital primarily hybridizes with Mn $e_g$ orbital ($d_{x^2-y^2}$ and $d_{z^2}$ orbitals)[32]. Considering the direct exchange interaction between Mn-Mn pairs and the Pauli exclusion principle, these pairs are likely to favor an antiferromagnetic configuration. However, due to the strong Coulomb interaction and weak crystal field strength between Mn atoms, and following Hund's rule, the $e_g$ orbital hosts 1–2 electrons. The intralayer Mn atoms form a Mn-Sn-Mn bond angle with their nearest Sn, approximately 60°, while the interlayer Mn atoms form a Mn-Sn-Mn bond angle of approximately 110°. The Mn atoms adopt a ferromagnetic configuration mediated by the unpaired Sn $p$ electrons, as supported by theoretical calculations shown in Fig. 3d. At the same

time, Mn atoms exhibit stronger metallic properties and are more susceptible to electron loss compared to Sn atoms. Hence, we propose a theoretical framework for the double exchange interaction, where electron transfer occurs between adjacent Mn cations of different valence states, mediated by Sn valence electrons, leading to ferromagnetic coupling. The double exchange mechanism occurs in two stages: Initially, a spin-down $p'$ electron from Sn is transferred to $Mn_1$ ion, turning $Mn^{2+}$ into $Mn^+$. Given the half-filled $d$ electron count in $Mn_2$ ions, the spin of the $p'$ electron couples antiparallel with the electron spins in $Mn_1$ ion; Subsequently, as the Sn $p''$ electron loses its paired electron, it participates in an indirect exchange interaction with $Mn_2$ ion. The $p''$ electron transitions to the $Mn_2$ ion, sharing a valence electron with aligned spins, thus converting $Mn^{3+}$ to $Mn^{2+}$. However, such a transition state is unstable, the initially formed $Mn^+$ easily loses an electron to revert to $Mn^{2+}$ ion, which then re-couples with Sn $p$ orbital, perpetuating the cycle. This completes the valence exchange between $Mn^{2+}$ and $Mn^{3+}$ ions, forming ferromagnetic coupling between Mn atoms mediated by Sn as the bridging atom.

Additionally, the Campbell model suggests that within the $3d$-$4f$ hybridization, there exists no direct exchange interaction between the rare-earth element Gd and the transition metal Mn. Instead, it is mediated by Gd $5d$ electrons[33,34]. The $4f$ electrons of the rare-earth element align parallel to the adjacent $5d$ electrons through RKKY interaction. Prior to the hybridization of Mn $3d$ and Gd $5d$ electrons, the $5d$ band is largely unoccupied, while the $3d$ band is half-filled, leading the $5d$ electrons to assume an antiparallel polarization relative to the $3d$ electrons, as depicted in the schematic diagram in Fig. 4c[35]. Thus, we have developed a model for the Mn-Mn double exchange interaction mediated by Sn atoms in $GdMn_6Sn_6$ through theoretical calculations. Highlighting Sn acts a crucial role in the spin frustration with kagome-Mn and the long-range magnetic order formation among magnetic anisotropic $4f$ electrons.

## Conclusions

To conclude, we conducted a comprehensive investigation of $GdMn_6Sn_6$ magnetic and electronic structure by using photoemission spectroscopy and DFT calculations. Our study demonstrated that localized kagome flat band is primarily attributed to the Mn-$e_g$ orbitals. Our experiments revealed the ferrimagnetic structure of $GdMn_6Sn_6$ by element-resolved XMCD and detected the magnetic moments in the nonmagnetic element Sn. Theoretical calculations highlighted double exchange interaction between Sn $5p$ and Mn $3d$ electrons, leading to the ferromagnetic coupling within Mn-Mn atoms, and the interlayer Gd-Mn atoms exhibit antiferromagnetic coupling resulting from the indirect interaction between rare-earth $f$ electrons and Mn $d$ electrons. Moreover, the orbital-resolved band structure calculations confirm the hybridization between Mn $3d$ and Sn $5p$ orbitals that plays a critical role in the magnetic coupling of $GdMn_6Sn_6$, which has been demonstrated in a similar material, $GdV_6Sn_6$, by resonant ARPES. Future experiments could provide more direct evidence of orbital hybridization in the $RMn_6Sn_6$ family. These results provide possibilities to explore the origin of the microscopic magnetic mechanism in the kagome magnets and offer potential connections to magnetic and topological properties in QAHE complex quantum systems. Furthermore, our research provides theoretical support and experimental evidence for advancements in applications such as spintronic devices and quantum computing, with the potential to drive the development of these technologies.

## Methods
### Crystal growth

High-quality single crystals of $GdMn_6Sn_6$ were synthesized using the flux method with tin as flux[10]. The starting elements Gd, Mn, and Sn, in a ratio of 1: 6: 20 and with a purity exceeding 99.9%, were placed in an alumina crucible within an evacuated quartz tube. The mixture was gradually heated to 1100 °C over an 8-h period, maintained at this temperature for 6 h, and then slowly cooled to 550 °C at a rate of 3 °C/h. Finally, the excess Sn flux was removed using a centrifuge. The optical microscopy image of the single

crystal and the crystallographic characterization are shown in Supplementary Note 1.

### ARPES experiments

The ARPES was performed at the BL03U beamline of the Shanghai Synchrotron Radiation Facility (SSRF) with a hemispherical electron-energy analyzer (Scienta-Omicron DA30) at a temperature of 16 K and using synchrotron light sources. The energy resolution of the ARPES was set at 20 meV, and the angular resolution was maintained at 0.2°. Samples were cleaved in situ, ensuring that the vacuum was better than $10^{-8}$ Pa during the whole measurements.

### XAS/XMCD experiments

XMCD reveals the differences in the absorption rates of left- and right-circularly polarized x-ray photons due to selection rules. Since the energy depends on specific elements, XMCD achieves element selectivity by choosing an appropriate photon energy range. In these experiments, the effect of left- and right- circularly polarized light was emulated by switching the direction of the magnetic field. The intensity of the photocurrent is measured to characterize the absorption rate of x-ray under positive and negative magnetic field, which namely the TEY (Total Electron Yield) method. Due to the shallow penetration depth of soft x-ray photons, the probing depth of the TEY method is limited to a few tens of nanometers below the surface. This work was conducted at the soft x-ray beamline BL07U of SSRF. All samples were cleaved in situ under a high (better than $10^{-5}$ Pa) vacuum and subsequently transferred to the measurement chamber (better than $10^{-6}$ Pa) equipped with eight vector magnets, capable of reaching a maximum magnetic field of 0.8 T. To eliminate systematic machine errors, all data were measured multiple times and averaged.

### The first-principles calculations

Electronic band structure calculations were performed within the framework of density functional theory (DFT) using the Vienna ab initio simulation package (VASP)[36], employing the projector augmented wave (PAW) method. The calculations were also performed using a self-consistent Green function method as it is implemented within the multiple scattering theory[29]. Spin-orbit coupling (SOC) effects have been included in the calculations. The generalized gradient approximation (GGA) of the Perdew-Burke-Ernzerhof (PBE) method was used for the exchange-correlation functionals[37]. Additionally, SOC effect was incorporated in the calculations. the spin-orbit coupling (SOC) effect was incorporated in the calculations. The kinetic energy cutoff for the plane-wave basis was set at 380 eV, and a $12 \times 12 \times 6$ $k$-mesh was selected with an accuracy of $10^{-8}$ eV in the calculations. The Fermi surface was calculated by constructing localized Wannier functions with the WANNIERTOOLS package[38]. To take into account strong correlation effects, we used a GGA + U density functional applied for Mn $3d$ states[39,40]. The value of the effective Hubbard parameter $U^*_{Mn} = U - J = 4 - 1 = 3$ eV was chosen to reproduce correctly Curie temperature (the same effective $U^*$ was used in a previous study[41,42]. As well, the GGA + U functional was used for treatment of strongly correlated Gd $4f$ states ($U^*_{Gd} = 6$ eV).

### Data availability

All data related to this paper, including raw data presented as Supplementary Data 1, are available from the corresponding author upon reasonable request.

### Code availability

The code used in this study is available from the corresponding author upon reasonable request.

**Article**

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

## Acknowledgements

This work was supported by the Science and Technology Commission of Shanghai Municipality (STCSM) (Grant No. 22560780300). This works was also partially supported by the National Key R&D Program of China (Grant No. 2022YFB3608000), the National Natural Science Foundation of China (Grant Nos. U1632266, 11927807, and U2032207), and Shanghai-XFEL Beamline Project (SBP). The ARPES experiments were performed with the approval of the Proposal Assessing Committee of $SiP.ME^2$ platform project (Proposal No. 11227902) supported by the National Science Foundation of China. A.E. acknowledges funding by Fonds zur Förderung der Wissenschaftlichen Forschung (FWF) Grant No. I 5384. We thank the staff from BL07U beamline of Shanghai Synchrotron Radiation Facility (SSRF) for the assistance of XMCD/XAS data collection. We acknowledge Q.S. Wu for the development of the user-oriented WANNIERTOOLS package.

## Author contributions

J. M. Wang and Q. Jiang performed the XMCD experiments with support from M. Ye, J. F. Cao and F. Y. Zhu; A. Ernst, V. N. Antonov and J. M. Wang performed the theoretical calculation; J. M. Wang synthesized the single crystal samples; Q. Jiang, H. J. Qian and D. Y. Wang provided assistance in the photoemission experiment; M. Ye and S. Qiao supervised and designed the experiments.

## Competing interests

The authors declare no competing interests.
