## [Transparent Peer Review file · Communications Physics]

Double Exchange Interaction in Mn-based Topological Kagome Ferrimagnet

Corresponding Author: Dr Mao Ye

Version 0:

Reviewer comments:

Reviewer #1

(Remarks to the Author)

In this paper, Wang et al. claim to thoroughly investigate the electronic structure and magnetic properties both experimentally (using element-resolved x-ray magnetic circular dichroism (XMCD) and angle-resolved photoemission (ARPES)) and theoretically (using density functional theory (DFT)). They also claim to have discovered induced magnetic moments in the nonmagnetic element Sn experimentally. However, there are several issues in this manuscript that need to be addressed before I can recommend it for publication:

Band Structure (ARPES and DFT):

1. Most Kagome lattice compounds exhibit a very similar electronic band structure (e.g., Dirac cones, topological flat bands, van Hove singularities). For example, see Scientific Reports 6, 25988 for Kagome magnets, or Phys. Rev. B 108, 045132 for RMn_6Sn_6 . What is the novelty in the band structure of GdMn_6Sn_6 ? How does this band structure contribute to the story of this manuscript?
2. It is unclear whether spin-orbit coupling (SOC) is included in the DFT results. The authors mention in the caption of Fig. 1 that SOC is not included, but in the methods section, they state that SOC is incorporated. What is the difference in the band structure with and without SOC?
3. To adjust the 4f bands for Gd, the authors use $U_{\text{eff}} = U - J = 6$ eV. What are the individual values of U and J? In transition elements, J is not crucial (we usually set $J = 0$), but in rare earth elements, a non-zero J is expected for a larger orbital moment, which makes the DFT calculations more reliable. Have the authors verified the occupation is correct in the DFT results (e.g., ensuring the density matrix is fully diagonalized)? The reliability of the calculated magnetic moments for Mn and Sn is questionable without this verification. Furthermore, the magnetic moments of Sn are very small (<0.2), which does not convincingly demonstrate an induced magnetic moment in Sn.

XMCD Results:

1. The authors mention that the x-ray absorption spectroscopy (XAS) intensity for Sn is relatively weak, but the intensity in XMCD for Sn is three orders of magnitude smaller, while the width of the peak is about five times broader compared to that for Gd and Mn. This result is not convincing evidence of an induced magnetic moment in Sn.
2. The fitting of experimental and theoretical results in Fig. 2c (XMCD) is not "quite well" to me. The fitting for Gd and Mn shows some mismatches but is overall acceptable, while the fitting for Sn is quite poor.
3. There are established methods to induce magnetic moments in nonmagnetic elements, such as the spin Hall effect. From my understanding, the role of Sn is more akin to mediating superexchange interactions (as shown in the schematic illustrations in Fig. 4d). The authors should clarify their claim by combining this result with DFT.

Exchange Interaction and Magnetic Moment:

1. The magnetic moment and magnetic structure of RMn_6Sn_6 have been discussed in several previous works (e.g., Rep. Prog. Phys. 86, 114502). It would be beneficial if the authors could enhance the novelty of their work.
2. Fig. 3 looks promising, but it is derived from the Heisenberg model, which has limitations in describing rare earth superexchange interactions and some transition metals. Additionally, the Heisenberg model assumes rotational symmetry, which is not the case for Kitaev magnets. It would be better if the authors could consider the Kitaev-Heisenberg model and discuss its limitations.

To conclude, the authors should clarify their motivation, highlight the breadth and potential of their research, and emphasize

the impact of their paper to merit publication in Communications Physics.

Reviewer #2

(Remarks to the Author)

The manuscript examines GdMn₆Sn₆ utilizing techniques such as XAS, XMCD, ARPES, and first-principles calculations. The main claim is that GdMn₆Sn₆ exhibits a ferrimagnetic structure and, notably, suggests that the typically nonmagnetic Sn atoms possess magnetic components. Furthermore, the authors propose the occurrence of a double exchange interaction involving Sn atoms. Given that the presence of a magnetic moment in Sn is highly unusual, this finding would indeed be notable if corroborated. I would recommend publication in Communications Physics, if the authors could address the following questions/comments.

- The authors mention the presence of a flat band within the range of 0.4-0.6 eV below the Fermi level (E_F) from the ARPES results, but this is not readily apparent to me. In fact, the bands seem to broaden and lack clear resolution in this energy region.
- To support the authors' central claim, it is crucial to observe features indicative of orbital mixing in the ARPES data, rather than simply identifying Dirac crossings or flat bands. Therefore, my question is whether the authors have identified any such orbital mixing features in their ARPES data that align with the findings from DFT calculations.
- Regarding the XMCD analysis,
 - 1) Line 156: the authors mention that the Sn1 site is nearest to the Mn Kagome layer. However, according to Fig. 2(d), it appears that the Sn2 site is actually closer. Could the authors please clarify this point?
 - 2) In Fig. 2(c), the Sn1 site appears notably different from the Sn3 site. Despite this, the authors describe the spin features of these sites as being similar. Could the authors provide a more detailed explanation of this claim?
- Given the proximity of the Sn3 site to the Gd atoms, is there a particular reason why the Mn site appears to impact the magnetism of Sn, while the adjacent Gd site does not seem to affect the Sn site in a similar manner?
- The readability of Fig. 3 presents some challenges. Firstly, the labels for Mn1, Mn2, and Mn3 in part (a) are virtually indistinguishable due to the small font size. Additionally, the negative lattice length used in the figure is confusing. What does this signify? Furthermore, the background repeating unit cells with color map only at the center could potentially be misleading.

Reviewer #3

(Remarks to the Author)

In this manuscript, Wang et al. study the magnetic interaction and electronic structure in the kagome magnet GdMn₆Sn₆ using both experimental probes and theoretical calculations. The authors claim a flat band originates from the Mn kagome layer and by a combination of ARPES and DFT. Furthermore, the authors use XAS and XMCD, which are consistent with the DFT calculations, to support the ferrimagnetism in this system. Consequently, given the weak moment exhibited on Sn sites, the authors try to explain the ferromagnetism in Mn atoms with exchange spins via Sn. However, the authors want to claim too many things in this work, with each of them not being strongly supported by their data.

First, although flat band is of particular interest in kagome lattice, the ARPES results shown in Fig. 1e cannot be "clear" evidence to support the existence of a flat band in GdMn₆Sn₆. I cannot really discern the flat band signature in the experimental data. Can the author show more EDCs to support the claim?

In addition, Fig. 1d is calculated based on "major/minor spin". However, there is no explanation in Line 114 when the authors start discussing this part. Also, the authors don't separate the orbital contributions in the calculations. It is hard to tell if the "flat band" at -0.5 eV comes from Mn d orbitals. Another related question is why the flat band in Fig. 1c is above E_F , but the DFT shows a flat band far below E_F in Fig. 1d. This seems inconsistent.

Second, the XAS and XMCD experimental data in Fig. 2 are quite confusing to me. Can the authors provide more experimental details? These are important for readers who are not familiar with these experimental probes. For example, why are I₊ and I₋ almost identical in XAS? What is the magnetic field direction? What is the meaning of solid and dashed arrows in the inset for +-B?

How do the authors obtain XMCD data? I can only see one curve, such as the green circles in Fig. 2a (bottom). Is it for +0.4 T or -0.4T? I'm also interested in the field dependence of the peak intensity by sweeping the magnetic field. Can the authors provide these results? The peak positions at Mn L₃ and Sn M₄ are not consistent between XAS and XMCD data. Can the authors explain why?

In Line 156, the authors state that Sn1 is the closest to the Mn layer, which seems different from what is shown in Fig. 2d. The Sn2 is the closest. Is this a mistake? Otherwise, the authors need to check the calculation results. The final theoretical proposal of the double exchange mechanism is based on the observation of weak magnetism on Sn in XMCD and the DFT calculation.

Third, how do the authors add the Coulomb interaction U? Can the authors display it in the Hamiltonian? The authors' claim about a C₂ symmetry-breaking distribution after adding U is quite questionable. From my view, the distributions in Fig. 3a-c are already of two-fold symmetry. The U term seems more like altering the interaction from negative to positive.

On the other hand, I feel like the ARPES results are not quite relevant to the argument about magnetism and the related origins in this kagome magnet. The authors may consider either making a stronger connection between the two parts or removing the ARPES data. But anyway, I don't think the current data supports the existence of a flat band.

Therefore, given these concerns, unfortunately, I cannot recommend the publication in Communications Physics at this point.

Additionally, I have some minor questions and suggestions:

1. What is the meaning of the sentence in Lines 65-67?

2. What is the meaning of "cruising magnetism" in Line 102?
3. Fig.1(d,e) are too small compared to 1a. Figure 1 should be rearranged.
4. The 3D view in Fig.1c is not a good example to show the Dirac point at the K point and vHS near the M point in the band structure. I suggest using the simpler version.
5. The inset in Fig. 2a is not clear in showing the experimental setup as depicted in Line 128.
6. Fig. 2d is not clear to show the moment direction. It would be great to show some views along the c-axis to display the orientations of the moments in each layer.
7. Line 96, c should be subscripted, like T_c
8. Line 127 has a typo: "magnetize"
9. Add references for Lines 137-138.
10. Line 141 has a typo: "non-magnetic"
11. Units are missing in Table 1.

Version 1:

Reviewer comments:

Reviewer #1

(Remarks to the Author)

The authors have made their explanation to all reviewers' comments, mostly convincing, also highlighted the significance of their work by serving half-filled 4f band Gd as an ideal model of studying the magnetic exchange interaction in RMn_6Sn_6 . The authors also improved their data quality (ARPES, etc.), clarified the importance of flat band signature in magnetic coupling magnetism, and provided a more detailed discussion of different sites Sn. The statement in the revised manuscript is more clear to me. Therefore, I would recommend this manuscript to publish in Communications Physics.

Reviewer #2

(Remarks to the Author)

The authors addressed the questions and comments from the referees well. However, one concern remains: the ARPES data does not support their main claim. Why not highlight the hybridization of the V case at the end of the paper and suggest possible connections and future studies?

Reviewer #3

(Remarks to the Author)

I appreciate the authors' detailed explanation, especially regarding the XMCD and XAS data. The ARPES data appears much clearer than in the previous version. The revised manuscript has been significantly improved in terms of clarity. The magnetic moment on the Sn site induced by double exchange interaction could be of potential interest to the community.

However, I noticed that the references are incorrectly sorted in the revised version. For example, in Line 21, the references are listed as 1-7, followed by 39. Additionally, in Line 53, Refs. 33 and 34 are theory papers, but the authors cite them as experimental evidence in response to Referee 1's question. The authors should carefully check the manuscript. I can recommend publication after these minor corrections.

RESPONSES TO REVIEWERS' COMMENTS:

To Reviewer #1:

In this paper, Wang et al. claim to thoroughly investigate the electronic structure and magnetic properties both experimentally (using element-resolved x-ray magnetic circular dichroism (XMCD) and angle-resolved photoemission (ARPES)) and theoretically (using density functional theory (DFT)). They also claim to have discovered induced magnetic moments in the nonmagnetic element Sn experimentally. However, there are several issues in this manuscript that need to be addressed before I can recommend it for publication:

Response: We sincerely thank the referee for his/her careful review and evaluation of our manuscript. We have carefully considered all the suggestions and comments, responded to the relevant questions, and revised the manuscript accordingly. Firstly, we have improved the introduction section to highlight the significance of our work in exploring the magnetic exchange interactions within RMn_6Sn_6 . We have also detailed the parameters of our theoretical calculations to ensure their reasonableness and consistency with the experimental results. Below are our specific responses to each comment:

Band Structure (ARPES and DFT):

Comment 1: *Most Kagome lattice compounds exhibit a very similar electronic band structure (e.g., Dirac cones, topological flat bands, van Hove singularities). For example, see Scientific Reports 6, 25988 for Kagome magnets, or Phys. Rev. B 108, 045132 for RMn_6Sn_6 . What is the novelty in the band structure of $GdMn_6Sn_6$? How does this band structure contribute to the story of this manuscript?*

Response: We thank the referee for the constructive suggestion to improve the statement on the importance of our work. To better highlight the novelty of the present study, we have revised the introduction section. As the common feature of the RMn_6Sn_6 materials, the band structure of the influenced by both the kagome lattice and magnetic properties. The flat band resulting from the geometric frustration of the Mn-Kagome layer exhibit highly localized electronic states. The different rare earth (R) ions generate exotic topological states through their complex exchange interactions. In the $GdMn_6Sn_6$, due to the half-filled $4f$ electron shell of Gd^{3+} , which results in zero orbital magnetic moment, it serves as an ideal model for studying the magnetic exchange interactions within the RMn_6Sn_6 . The magnetic ordering originates from the unique electronic states of different materials. Therefore, exploring the band structure of $GdMn_6Sn_6$ is significant for researching the interactions between rare earth $4f$ electrons and transition metal $3d$ electrons. It also aids in understanding the complex magnetic behaviors and electronic transport properties within these materials. In previous studies, such as the impact of different electron doping levels on the topological properties of kagome Mott insulators (*Scientific Reports* **6**, 25988), and the magnetic anisotropy and topological band properties of the RMn_6Sn_6 (*Phys. Rev. B* **108**, 045132), these issues have been thoroughly discussed. However, the investigation on the mechanism of magnetic properties, especially the exchange interactions between R -Mn and the magnetic coupling of RMn_6Sn_6 remain are still lacking⁴⁰, which prevent the further tuning of the Curie temperature and magnetic anisotropy that facilitate the practical applications in spintronic devices and quantum computing.

Therefore, our present study aims to explore the band structure of GdMn_6Sn_6 and investigate the underlying mechanism between electronic states and magnetic ordering, which reveals the magnetic double exchange interactions mediated by Sn.

=>Revised manuscript: *Line 47-61*: More importantly, the establishment of long-range magnetic ordering requires the itinerant magnetism of valence band electrons, and the magnetic order is closely related to the unique highly localized flat band generated by the Mn-kagome geometric lattice in RMn_6Sn_6 . This localization competes with the itinerant electrons in the formation of the magnetic ordered ground states in GdMn_6Sn_6 , making the investigation of the electronic states particularly important. Although neutron diffraction experiments have revealed ferromagnetic coupling within the kagome-Mn layer^{33,34}, the relationship between these electronic states and magnetism remains unclear. Furthermore, the magnetic properties such as the magnetic anisotropy, are also significantly influenced by the rare-earth elements and the coupling between Mn and rare-earth layers, indicating that the $4f$ electrons of the rare earth elements also participate in the magnetically ordered coupling. However, the investigation on the mechanism of magnetic properties, especially the exchange interactions between R -Mn and the magnetic coupling of RMn_6Sn_6 remain are still lacking from either experimental or theoretical aspects⁴⁰, which is critical for investigating their topological properties and the development of applications based on magnetic kagome materials.

Comment 2: *It is unclear whether spin-orbit coupling (SOC) is included in the DFT results. The authors mention in the caption of Fig. 1 that SOC is not included, but in the methods section, they state that SOC is incorporated. What is the difference in the band structure with and without SOC?*

Response: We apologize for the confusing description on the calculation method regarding the spin-orbit coupling (SOC) in our density functional theory (DFT) results. In fact, we have performed calculations both with and without SOC. Given that the SOC effect in GdMn_6Sn_6 is relatively small and has a very limited impact on the band structure, we chose to present the results without SOC in the original manuscript to highlight the primary electronic characteristics without introducing the complexities brought about by SOC. In the original manuscript, the tight-binding model calculations in Fig. 1(c) do not include SOC and only consider nearest-neighbor electron hopping. This is merely to illustrate the theoretical band structure. To be consistent in the revised manuscript, we will show and discuss only full-relativistic calculations that account for SOC in the revised manuscript. SOC can induce a 16 meV gap at the Dirac point at the K point, with subtle changes in the bands near the Fermi level (see Supplementary Materials).

=>Revised manuscript (see Supplementary Materials):

Comment 3: *To adjust the 4f bands for Gd, the authors use $U_{eff} = U - J = 6$ eV. What are the individual values of U and J ? In transition elements, J is not crucial (we usually set $J = 0$), but in rare earth elements, a non-zero J is expected for a larger orbital moment, which makes the DFT calculations more reliable. Have the authors verified the occupation is correct in the DFT results (e.g., ensuring the density matrix is fully diagonalized)? The reliability of the calculated magnetic moments for Mn and Sn is questionable without this verification. Furthermore, the magnetic moments of Sn are very small.*

Response: In our calculations we have used two different values for U and J . Since the total orbital moment of Gd ion is zero, the value of J is not crucial. $U_{eff} = 7 - 1 = 6$ eV and $U_{eff} = 6 - 0 = 6$ eV provide very similar band structures. The choice of U_{eff} for Gd was made to adjust positions of Gd 4f states in accordance with known photoemission data: all 7 4f electrons of Gd in spin up channel are occupied and strongly localized. Similar results for the valence band structure were obtained using a self-interaction correction, which pushes 4f Gd states far below the Fermi level. To be consistent we discuss only results obtained with a GGA+U method, which was also applied for Mn 3d states ($U_{eff} = 4 - 1 = 3$ eV). This choice for Mn 3d states was referred to obtain magnetic order, magnetic moments for Mn and Curie temperature in agreement with experiment (*Phys. Rev. B* **108**, 045132 for RMn_6Sn_6). We observed ferromagnetic alignment in the in-plane Mn-kagome layers. Additionally, the Curie temperature was estimated to be 410 K using the Heisenberg model within a random phase approximation utilizing exchange parameters obtained with the magnetic force theorem. Regarding the diagonalization of the density matrix, we ensured that the density matrix was fully diagonalized during the calculations. This guarantees that our DFT results are accurate and reliable. As for the magnetic moment of Sn, it is an induced magnetic moment. Consequently, both the calculated and experimental magnetic moments are very small, which aligns with our expectations.

=>Revised manuscript: *Line 308-313:* To take into account strong correlation effects, we used a GGA+U density functional applied for Mn 3d states^{27,28,37}. The value of the effective Hubbard parameter $U_{Mn}^* = 4 - 1 = 3$ eV was chosen to reproduce correctly Curie temperature (the same effective U^* was used in a previous study^{17,40}). As well, the GGA+U functional was used for treatment of strongly correlated Gd 4f states ($U_{Gd}^* = 6$ eV).

XMCD Results:

Comment 4: *The authors mention that the x-ray absorption spectroscopy (XAS) intensity for Sn is relatively weak, but the intensity in XMCD for Sn is three orders of magnitude smaller, while the width of the peak is about five times broader compared to that for Gd and Mn. This result is not convincing evidence of an induced magnetic moment in Sn.*

Response: The magnetic moment of Sn is an induced magnetic moment, resulting in low peak intensity in the XMCD spectra. On the other hand, since Sn is a main group element with more delocalized p electrons, this leads to broader absorption peaks (*ACS Applied Materials & Interfaces* **7**(22):12074-12079). Consequently, the spectra of XMCD that reflecting the induced magnetism on Sn also exhibit a relatively broad feature. The dichroism signal at Sn M_5 edge ($I^+ - I^- = 8.5 \times 10^{-3}$ as shown in Fig. 2(c)) is only 0.264% of the XAS intensity ($I^+ + I^- = 3.225$ estimated from the edge-jump of the Sn M_5 edge). Compared with theoretical calculations, the results also indicate that the weak XMCD signal at the $M_{4,5}$ edges originate from its induced intrinsic magnetism rather than any experimental asymmetry. Therefore, both our experimental and

theoretical results demonstrate the existence of Sn-induced magnetic moments.

Comment 5: *The fitting of experimental and theoretical results in Fig. 2c (XMCD) is not "quite well" to me. The fitting for Gd and Mn shows some mismatches but is overall acceptable, while the fitting for Sn is quite poor.*

Response: Our calculations are based on a simple one-electron model and cannot reproduce correctly many body effects. Simulations of L edge agree typically better with experiment, while for M edge first-principles calculation cannot reproduce multiples and other many body effects. However, the current calculations provide a very similar trend obtained in our experiment.

Comment 6: *There are established methods to induce magnetic moments in nonmagnetic elements, such as the spin Hall effect. From my understanding, the role of Sn is more akin to mediating superexchange interactions (as shown in the schematic illustrations in Fig. 4d). The authors should clarify their claim by combining this result with DFT.*

Response: The Sn atom indeed acts as a mediator in the magnetic interaction, specifically through double exchange interaction. Unlike super exchange interaction, double exchange involves electron transfer between metal ions of different valence states via anions. This transfer tends to align the spins of adjacent metal ions, leading to ferromagnetic ordering. In contrast, super exchange is an indirect orbital interaction mediated by anions. To demonstrate this, we can employ the Disordered Local Moment (DLM) method. The DLM method simulates a high-temperature disordered state where the induced magnetic moment of Sn no longer exists, and the orientations are random. By comparing the results of the system's ferrimagnetic state at T=0 K with those obtained using the DLM method, we can understand the influence of the induced magnetic moment on the electronic and magnetic properties. We will include this discussion in the revised manuscript to provide a deeper insight into the role of Sn in the magnetic interactions.

=>Revised manuscript: *Line 213-222:* Simultaneously, using the Disordered Local Moment (DLM) method to simulate the magnetic properties of GdMn_6Sn_6 in a high-temperature disordered state also verified the presence of indirect double exchange interaction. In the DLM model, thermal perturbations are introduced, assuming that the magnetic moments in the system are disordered, meaning that the direction of each atom's magnetic moment is random. This allows us to obtain the electronic structure and energy distribution in the disordered magnetic state. By comparing the total energy and atomic magnetic moment distribution at T=0 K with those calculated using the DLM method, it was found that the system tends to form a magnetically ordered state at low temperatures, which is consistent with the ferromagnetic ordering induced by double exchange interaction.

Exchange Interaction and Magnetic Moment:

Comment 7: *The magnetic moment and magnetic structure of RMn_6Sn_6 have been discussed in several previous works (e.g., *Rep. Prog. Phys.* **86**, 114502). It would be beneficial if the authors could enhance the novelty of their work.*

Response: We agree with the referee and extend our discussion in the revised manuscript accordingly. In the paper '*Rep. Prog. Phys.* **86**, 114502,' the magnetic properties and magnetic structures of RMn_6Sn_6 have been discussed in detail, elucidating how the magnetic ordering is influenced by R-Mn interactions. It also notes that the size of the topological Dirac gap varies with

different R atoms. However, the microscopic magnetic coupling mechanisms within RMn_6Sn_6 remain unclear. The highlight of our work is to reveal the exchange interactions in the $GdMn_6Sn_6$ system and their impact on magnetism. This research enriches the understanding of microscopic magnetic interaction mechanisms within RMn_6Sn_6 . To enhance the novelty of our present work, we have revised the manuscript as follow:

=>Revised manuscript: Line 47-61: More importantly, the establishment of long-range magnetic ordering requires the itinerant magnetism of valence band electrons, and the magnetic order is closely related to the unique highly localized flat band generated by the Mn-kagome geometric lattice in RMn_6Sn_6 . This localization competes with the itinerant electrons in the formation of the magnetic ordered ground states in $GdMn_6Sn_6$ materials, making the investigation of the electronic states particularly important. Although neutron diffraction experiments have revealed ferromagnetic coupling within the kagome-Mn layer^{33,34}, the relationship between these electronic states and magnetism remains unclear. Furthermore, the magnetic properties such as the magnetic anisotropy, are also significantly influenced by the rare-earth elements and the coupling between Mn and rare-earth layers, indicating that the $4f$ electrons of the rare earth elements also participate in the magnetically ordered coupling. However, the investigation on the mechanism of magnetic properties, especially the exchange interactions between R -Mn and the magnetic coupling of RMn_6Sn_6 remain are still lacking from either experimental or theoretical aspects⁴⁰, which is critical for investigating their topological properties and the development of applications based on magnetic kagome materials.

Line 270-272: Furthermore, our research provides significant theoretical support and experimental evidence for advancements in applications such as spintronic devices and quantum computing, with the potential to drive the development of these technologies.

Comment 8: *Fig. 3 looks promising, but it is derived from the Heisenberg model, which has limitations in describing rare earth superexchange interactions and some transition metals. Additionally, the Heisenberg model assumes rotational symmetry, which is not the case for Kitaev magnets. It would be better if the authors could consider the Kitaev-Heisenberg model and discuss its limitations.*

Response: A key quantity in our study is the exchange constant J , obtained from first-principles calculations, which illustrates the strength of these interactions. The Kitaev-Heisenberg model was originally proposed to describe complex magnetic interactions in certain heavy transition metal compounds with strong spin-orbit coupling (SOC), such as Ir or Ru based two-dimensional honeycomb lattice materials. In these materials, magnetic interactions include both the traditional Heisenberg interaction and the anisotropic Kitaev interaction induced by strong SOC. However, the highly localized nature of the $4f$ electrons, the weak spin-orbit coupling (SOC) effects, and the isotropic spin interactions between electrons within the Mn layers within the Mn layers for the $GdMn_6Sn_6$ make the use of the traditional Heisenberg model more appropriate (*Kitaev A. 321(1), 2-111(2006)*). In our research, we used the Heisenberg model exclusively to calculate the Curie temperature. When discussing the J values, we mapped the strength and distribution of magnetic exchange interactions in real space. This approach provides a comprehensive understanding of the exchange interactions' impact on the system's electronic and magnetic properties.

=>Revised manuscript: Line 159-163: Understanding the strength of interaction within the kagome lattice is essential for unraveling the magnetic properties of this material. Due to the weak SOC effect in $GdMn_6Sn_6$ and the isotropic exchange interactions within the Mn layers, we adopt

the Heisenberg model to describe the spin correlations instead of Kitaev model⁴³.

To Reviewer #2:

The manuscript examines GdMn₆Sn₆ utilizing techniques such as XAS, XMCD, ARPES, and first-principles calculations. The main claim is that GdMn₆Sn₆ exhibits a ferrimagnetic structure and, notably, suggests that the typically nonmagnetic Sn atoms possess magnetic components. Furthermore, the authors propose the occurrence of a double exchange interaction involving Sn atoms. Given that the presence of a magnetic moment in Sn is highly unusual, this finding would indeed be notable if corroborated. I would recommend publication in Communications Physics, if the authors could address the following questions/comments.

Response: We sincerely thank the referee for his/her careful review and positive evaluation of our manuscript. We have carefully considered all the suggestions and comments raised by the referee, and revised the manuscript accordingly. Especially, we have provided more comprehensive ARPES data to demonstrate the existence of the flat band and have revised the unclear schematic diagrams of the magnetic moment structure and exchange interactions in the manuscript. These changes enhance the readability and significance of our work. Below are our specific responses to each of your comments:

Comment 1: *The authors mention the presence of a flat band within the range of 0.4-0.6 eV below the Fermi level (EF) from the ARPES results, but this is not readily apparent to me. In fact, the bands seem to broaden and lack clear resolution in this energy region.*

Response: GdMn₆Sn₆ band structure in the original manuscript was indeed not clear enough, which is mainly attributed to the relatively high spectral background. To enhance the visibility of the flat band, we improved the presentation of the ARPES image by optimizing the color scale and showing the ARPES image in the wider momentum range. In this image, the flat band can be seen clearly in the 0.4-0.6 eV range. We also integrated the intensity along the momentum direction, showing an integrated energy distribution curve (EDC) as reference. The EDC along the Γ -K direction shows a distinct peak for the flat band. Additionally, through variable photon energy experiments, we have clearly observed the dispersion of the flat band along the k_z direction. These data are provided in the supplementary materials.

=>Revised manuscript:

Comment 2: *To support the authors' central claim, it is crucial to observe features indicative of*

orbital mixing in the ARPES data, rather than simply identifying Dirac crossings or flat bands. Therefore, my question is whether the authors have identified any such orbital mixing features in their ARPES data that align with the findings from DFT calculations.

Response: We totally agree that the evidence for orbital hybridization is crucial to support the central argument regarding magnetic exchange interactions. In present study, we revealed the induced magnetic moment on Sn, which is originated from the hybridization between Mn *d* and Sn *p* electrons. This is fully supported by our DFT calculation, where the orbital resolved band structure calculations of GdMn₆Sn₆ clearly show the band overlap with Mn *d* and Sn *p* electrons (see Supplementary Materials). Further verification on the orbital hybridization can be demonstrated through resonant ARPES experiments, as shown in related GdV₆Sn₆ material (*RRL 17(12) (2023): 2300083*), where resonant ARPES clearly observed the V 3*d* resonance orbitals overlapping with the energy bands of Sn. Combined with orbital-resolved DFT band calculations, it was found that the V 3*d* electrons hybridize with Sn 5*p* electrons indicating that the *d* electrons are not fully localized. This hybridization facilitates the formation of long-range magnetic ordering.

Regarding the XMCD analysis:

Comment 3: Line 156: the authors mention that the Sn1 site is nearest to the Mn Kagome layer. However, according to Fig. 2(d), it appears that the Sn2 site is actually closer. Could the authors please clarify this point?

Response: We thank for the careful examination and apologize for the confusing presentation on in Fig. 2(d). The actual spatial distances between Mn and Sn1 and Sn2 atoms are 2.71 Å and 2.89 Å, respectively. The false impression generated in Fig. 2(d) is attributed view angle of the structural image. To eliminate the such misleading impression, we redraw the lattice structure diagram and make appropriate modifications and explanations in the manuscript. We believe that the revised manuscript and graph can correctly show that the Sn1 site is closer to Mn atoms and more likely to participate in exchange interactions (*Riberolles, S. X. M., et al. PRX 12.2 (2022): 021043*).

=>**Revised manuscript:** Line 148-153: In a unit cell, the Sn1, Sn2, and Sn3 atoms are each in different chemical environments. The Sn1 site is sandwiched between the upper and lower Mn kagome layers, allowing it to directly participate in the Mn-Sn-Mn magnetic exchange interaction⁴⁴. It is also spatially closer to the Mn atoms, resulting in the largest induced magnetic moment. Although the Sn2 site is close to the Mn kagome layer in the *c* direction, it is pushed to the outside of the unit cell by the Gd atoms, thus reducing its induced magnetic moment.

Comment 4: In Fig. 2(c), the Sn1 site appears notably different from the Sn3 site. Despite this, the authors describe the spin features of these sites as being similar. Could the authors provide a more

detailed explanation of this claim?

Response: We apologize for not providing a detailed discussion of the calculated site-resolved spectra for the Sn elements at the three different sites in the original manuscript, which may have caused confusion for the readers. We have redrawn the schematic diagram of the magnetic structure to correspond more clearly with the calculation results. Sn1, Sn2, and Sn3 are in different chemical environments, resulting in variations in the induced magnetic moments. Within a single unit cell, the Sn1 and Sn3 layers are vertically positioned close to the Mn layer, with Sn1 being slightly closer. Sn1 is located between two ferromagnetic kagome Mn layers, allowing it to directly participate in Mn-Sn-Mn exchange interactions, which induces a relatively larger magnetic moment. However, Sn3 atoms are situated in the plane of the Gd layer and are influenced by the Gd layer electrons, which diminishes their induced magnetic moment. Consequently, the XMCD results for Sn1 and Sn3 sites are different.

=>Revised manuscript: *Line 148-154:* In a unit cell, the Sn1, Sn2, and Sn3 atoms are each in different chemical environments. The Sn1 site is sandwiched between the upper and lower Mn kagome layers, allowing it to directly participate in the Mn-Sn-Mn magnetic exchange interaction⁴⁴. It is also spatially closer to the Mn atoms, resulting in the largest induced magnetic moment. Although the Sn2 site is close to the Mn kagome layer in the c direction, it is pushed to the outside of the unit cell by the Gd atoms, thus reducing its induced magnetic moment. The Sn3 atoms are influenced by the Gd atoms, which weakens their induced magnetic moments.

Comment 5: *Given the proximity of the Sn3 site to the Gd atoms, is there a particular reason why the Mn site appears to impact the magnetism of Sn, while the adjacent Gd site does not seem to affect the Sn site in a similar manner?*

Response: As shown in Fig. 2(e) in the revised manuscript, within the kagome-Mn layer, the distance between a Mn atom and the Gd and Sn3 atoms is 3.20 Å and 2.76 Å, respectively. Due to the closer proximity of Mn to Sn3, the interaction between Mn and Sn3 is stronger. In contrast, the neighboring Gd atoms have highly localized 4f electrons, which do not easily hybridize with the Sn p electrons.

Comment 6: *The readability of Fig. 3 presents some challenges. Firstly, the labels for Mn1, Mn2, and Mn3 in part (a) are virtually indistinguishable due to the small font size. Additionally, the negative lattice length used in the figure is confusing. What does this signify? Furthermore, the background repeating unit cells with color map only at the center could potentially be misleading.*

Response: We thank the referee for the constructive suggestion for the improvement on the readability of Fig. 3. For better readability, firstly, we adjusted the font size and contrast of Fig. 3. Then we add detailed explanation about the graphs presented in Fig.3 in the revised manuscript. Fig. 3 illustrates the exchange interaction strengths between a given Mn atom (the Mn atom at the center) and its nearest neighbors, including in-plane kagome-Mn, interlayer Mn-Mn, and interlayer Gd-Mn atoms. Red (blue) reign indicates positive (negative) J values, representing ferromagnetic (antiferromagnetic) coupling. The lines in the figure represent the contour lines of J values (meV), and the negative lattice represents the relative lattice lengths in real space from the central atom, making the interaction strengths more intuitive. From Fig. 3, it is evident that the exchange interaction strength is strongest within the unit cell and decreases sharply beyond the distance of two atoms, which is why the colormap intensity is highest at the center.

=>Revised manuscript: *Line 171-176:* Fig. 3 shows the exchange interaction strength J (in meV)

between intralayer and interlayer Mn-Mn, Gd-Mn atoms in the real-space supercell, without and with the Hubbard U correction, respectively, where the axes (in unit of Å) indicate the relative lattice position in real space from the central atom. The color scale in red and blue represent positive (ferromagnetic coupling) and negative (antiferromagnetic coupling) values of J between this site atom with the central atom, respectively.

To Reviewer #3:

In this manuscript, Wang et al. study the magnetic interaction and electronic structure in the kagome magnet GdMn6Sn6 using both experimental probes and theoretical calculations. The authors claim a flat band originates from the Mn kagome layer and by a combination of ARPES and DFT. Furthermore, the authors use XAS and XMCD, which are consistent with the DFT calculations, to support the ferrimagnetism in this system. Consequently, given the weak moment exhibited on Sn sites, the authors try to explain the ferromagnetism in Mn atoms with exchange spins via Sn. However, the authors want to claim too many things in this work, with each of them not being strongly supported by their data.

Response: We sincerely thank the referee for his/her careful review and positive evaluation of our manuscript. In the revised manuscript, we have carefully considered all the suggestions and comments raised by the referee, and made modifications and improvements accordingly. We have optimized presentation of the ARPES data in the manuscript to clearly distinguish the flat band and explain its significance in exploring the magnetic coupling mechanisms of the material. Additionally, we have provided more detailed supplementary information on the XMCD experiments. We response to all the comments in a point-to-point way as follow.

Comment 1: *First, although flat band is of particular interest in kagome lattice, the ARPES results shown in Fig. 1e cannot be “clear” evidence to support the existence of a flat band in GdMn6Sn6. I cannot really discern the flat band signature in the experimental data. Can the author show more EDCs to support the claim?*

Response: The band structure of GdMn6Sn6 was not clear enough in the original manuscript, primarily due to a relatively high spectral background. To improve the visibility of the flat band, we have optimized the presentation of the ARPES image by adjusting the color scale and displaying the image over a wider momentum range. This adjustment allows the flat band to be clearly observed in the 0.4-0.6 eV range. Additionally, we have integrated the intensity along the momentum direction to produce an integrated energy distribution curve (EDC) for reference. The EDC along the Γ -K direction reveals a distinct peak corresponding to the flat band. Moreover, variable photon energy experiments have enabled us to observe the dispersion of the flat band along the k_z direction. These data are included in the supplementary materials.

=>Revised manuscript:

Comment 2: Fig. 1d is calculated based on “major/minor spin”. However, there is no explanation in Line 114 when the authors start discussing this part. Also, the authors don’t separate the orbital contributions in the calculations. It is hard to tell if the “flat band” at -0.5 eV comes from Mn *d* orbitals. Another related question is why the flat band in Fig. 1c is above E_F , but the DFT shows a flat band far below E_F in Fig. 1d. This seems inconsistent.

Response: We thank the referee for his/her careful examination and apologize for the confusing description and data presentation in the manuscript. As pointed out by the referee, the discussion on the calculated spin-resolved band (Fig. 1d) is almost missing. Since there are no related spin-resolved experiments in present study to compare with, and its relation with the magnetic exchange interactions discussed in the paper is also weak, we have rearranged Fig. 1, removing the spin-resolved band structure. Additionally, we have shown the band structure with orbital resolution in the supplementary materials, demonstrating that the flat band is primarily contributed by the Mn *d* orbitals, with overlaps from the Sn *p* electrons.

Fig. 1(c) illustrates the band structure of the kagome lattice calculated using a tight-binding model. Our calculations adopt the single-electron approximation, considering only the hopping parameter *t* for electron transitions between nearest-neighbor atoms to understand the electronic behavior in the periodic kagome lattice. By solving the Hamiltonian of the system, we qualitatively identify the existence of flat bands and Dirac points in the kagome system (see Supplementary Materials). In practical research systems, more complex interactions need to be considered, such as SOC and other strong correlation effects (e.g., the Hubbard *U* term). These interactions significantly impact the band structure and the relative positions of the bands. As a strongly correlated system, the flat band in the GdMn₆Sn₆ system is located below the Fermi level.

Comment 3: The XAS and XMCD experimental data in Fig. 2 are quite confusing to me. Can the authors provide more experimental details? These are important for readers who are not familiar with these experimental probes. For example, why are I^+ and I^- almost identical in XAS? What is the magnetic field direction? What is the meaning of solid and dashed arrows in the inset for $+B$?

Response: We thank the referee for his/her constructive suggestion to enhance the clarity of the manuscript. We provided more detailed explanations of the XAS and XMCD experiments in the Methods section of the revised manuscript. The XMCD experiment is based on the principle of optical selection rules. The absorption intensity differs when the magnetization direction is parallel or anti-parallel with the vector of the circularly polarized x-ray, where the difference of the absorption intensity is the so-called magnetic circular dichroism.

Experimentally, the direction of the magnetic field can be altered while keeping the polarization

of the light constant (Parallel to the light path and anti-parallel to the light path) to obtain absorption spectra under different magnetic fields, thus obtaining the XMCD signal. In the XAS spectra, I^+ and I^- represent the absorption intensities of the sample under positive and negative magnetic fields, respectively. The intrinsic absorption spectra of the material are primarily determined by its elemental composition and electronic structure, which do not significantly change under different magnetic fields. Therefore, most of the absorption intensity remains consistent. In our XMCD experiments, the external magnetic field direction is always parallel to the light path. In the inset of Fig. 2(a), the $\pm B$ represents different magnetic field directions, which is equivalent to switching the polarization states.

We have improved the description on the XMCD methods and experimental setup in the revised manuscript as follow:

=>Revised manuscript: line 104-115: XMCD experiments are based on the principle of magnetic circular dichroism. Magnetic samples exhibit different absorption intensities under different polarized light due to optical selection rules. This phenomenon arises because polarized light breaks the symmetry of the system, leading to selective excitation of electronic states. Experimentally, switching the direction of the magnetic field and the direction of polarization is equivalent. The magnetic field changes the electronic states through Zeeman splitting, thereby producing a difference in the absorption spectra, which is used to analyze the magnetic properties of the material. Fig. 2(d) illustrates the experimental setup. The XAS spectra of the sample are obtained using circularly polarized light with positive (black) and negative (red) polarization or by applying an external magnetic field parallel (or antiparallel) to the direction of the light beam. The difference between these spectra constitutes the XMCD spectrum.

Line 286-298: XMCD reveals the differences in the absorption rates of left- and right-circularly polarized x-ray photons due to selection rules. Since the energy depends on specific elements, XMCD can achieve element selectivity by choosing an appropriate photon energy range. In these experiments, the effect of left and right circularly polarized light was emulated by switching the direction of the magnetic field. The intensity of the photocurrent is measured to characterize the absorption rate of x-ray under positive and negative magnetic field, which namely the TEY method. Due to the shallow penetration depth of soft x-ray photons, the probing depth of the TEY method is limited to a few tens of nanometers below the surface. This work was conducted at the soft x-ray beamline BL07U of SSRF. All samples were cleaved *in situ* under a high (better than 10^{-5} Pa) vacuum and subsequently transferred to the measurement chamber (better than 10^{-6} Pa) equipped with eight vector magnets, capable of reaching a maximum magnetic field of 0.8 T. To eliminate systematic machine errors, all data were measured multiple times and averaged.

Comment 4: *How do the authors obtain XMCD data? I can only see one curve, such as the green circles in Fig. 2a (bottom). Is it for +0.4 T or -0.4T? I'm also interested in the field dependence of the peak intensity by sweeping the magnetic field. Can the authors provide these results? The peak positions at Mn L3 and Sn M4 are not consistent between XAS and XMCD data. Can the authors explain why?*

Response: The XMCD spectra are extracted as the difference of the absorption spectra when the magnetization is applied parallel or anti-parallel with the vector of circularly polarized x-ray. Experimentally, two absorption spectra are obtained by switching the magnetization direction, while keeping the vector of x-ray beam fixed, resulting two absorption spectra indicated as I^- (blue) and I^+ (red) as shown in the upper panel of Fig. 2(a). The difference of I^- and I^+ (red) is then extracted

as XMCD signal, being shown as green circles in the lower panel of Fig. 2(a). In order to achieve better statistics (signal-to-noise ratio) of XMCD, we conducted multiple measurements and average the results for each XMCD spectrum. The same methods and operation are also applied for Mn and Sn absorption edges as shown in Fig. 2(b) and (c).

As for the field dependence of the peak intensity, once the material reaches saturation magnetization, the peak intensity of the XMCD spectra is independent of the magnetic field strength. We ensure that the measurements are conducted under sufficiently high magnetic fields (± 0.4 T) to achieve saturation (*G. Dhakal, et al. Physical Review B* **104.16: L161115 (2021)).**

The shift in peak positions between the XAS and XMCD spectra at the same absorption edge is mainly due to energy level splitting. As shown in the figure below, the Mn $L_{2,3}$ edges exhibit fine absorption spectra (*JESRP* **171.1-3: 24-29(2009)**), where the $L_{2,3}$ edges consist of multiple fine structure peaks. The peaks without magnetic contributions will disappear when XMCD is extracted from two absorption spectra (I^- and I^+), resulting in a shift between the XMCD and XAS peak positions.

Comment 5: In Line 156, the authors state that Sn1 is the closest to the Mn layer, which seems different from what is shown in Fig. 2d. The Sn2 is the closest. Is this a mistake? Otherwise, the authors need to check the calculation results. The final theoretical proposal of the double exchange mechanism is based on the observation of weak magnetism on Sn in XMCD and the DFT calculation.

Response: We appreciate the thorough review and apologize for the confusion presented in Fig. 2(d). The actual spatial distances between the Mn and Sn1 atoms and the Mn and Sn2 atoms are 2.71 Å and 2.89 Å, respectively. The misleading impression in Fig. 2(d) is due to the viewing angle of the structural image. To clarify this issue, we have redrawn the lattice structure diagram and made appropriate modifications and explanations in the manuscript. The Sn2 atoms are located on the outer side of the unit cell relative to the Mn atoms, and the actual spatial distance is larger than that of the Sn1 atoms. We believe that the revised manuscript and figure accurately show that the Sn1 site is closer to the Mn atoms and is more likely to participate in exchange interactions (*Riberolles, S. X. M., et al. PRX* **12.2: 021043(2022)**).

=>Revised manuscript: *Line 149-153:* The Sn1 site is sandwiched between the upper and lower Mn kagome layers, allowing it to directly participate in the Mn-Sn-Mn magnetic exchange interaction⁴⁴. It is also spatially closer to the Mn atoms, resulting in the largest induced magnetic moment. Although the Sn2 site is close to the Mn kagome layer in the c direction, it is pushed to the outside of the unit cell by the Gd atoms, thus reducing its induced magnetic moment.

Comment 6: How do the authors add the Coulomb interaction U ? Can the authors display it in the Hamiltonian? The authors' claim about a C_2 symmetry-breaking distribution after adding U is quite questionable. From my view, the distributions in Fig. 3a-c are already of two-fold symmetry. The U term seems more like altering the interaction from negative to positive.

Response: In strongly correlated systems, the Coulomb interaction U plays a crucial role. In our calculations, we introduce effective Coulomb interactions U_{eff} of 6 eV and 3 eV for Gd and Mn, respectively. When incorporating the Coulomb interaction U into this system, it is necessary to extend the Hamiltonian using the Hubbard model, where $n_{i\sigma}$ is the number operator for electrons with spin σ at site i .

$$H = -J \sum_{i,j} J_{i,j} \mathbf{S}_i \cdot \mathbf{S}_j + U \sum_i n_{i\uparrow} n_{i\downarrow}$$

Regarding the symmetry issue of U , we apologize for our misstatement in our manuscript. The interaction strength diagrams, both considering U and not considering U , exhibit twofold symmetry. The presence of the U term causes the Mn-Mn atomic interactions to shift from antiferromagnetic coupling to ferromagnetic coupling based on double exchange interactions. In the revised manuscript, we modified relevant sentences as follow:

=>**Revised manuscript:** Line 184-189: Notably, the Coulomb interaction U causes the exchange integral strength between Mn atoms to shift from negative to positive due to the charge transfer between Mn atoms, while the distribution of J maintains C_2 symmetry. Electrons with the same spin transition between different energy levels among the Mn atoms, leading to ferromagnetic coupling between Mn atoms within the kagome layer and reducing the absolute value of the exchange interaction strength.

Comment 7: I feel like the ARPES results are not quite relevant to the argument about magnetism and the related origins in this kagome magnet. The authors may consider either making a stronger connection between the two parts or removing the ARPES data. But anyway, I don't think the current data supports the existence of a flat band.

Response: We thank the referee for the constructive suggestion. In the revised manuscript, in order to clarify the relevance of the ARPES data with the main topic of this study, we have modified the introduction section. The magnetic ordering of the system originates from the unique electronic states of the material. Therefore, exploring the band structure of $GdMn_6Sn_6$ is significant for studying the interactions between rare-earth $4f$ electrons and transition metal $3d$ electrons, as well as for understanding the complex magnetic behaviors and electronic transport properties within this system. The kagome lattice structure in this system leads to the presence of flat bands, which exhibit

very strong localization. Simultaneously, the $4f$ electrons of Gd are also highly localized, resulting possible competition with the formation of long-range magnetic order of the system. By analyzing the orbital composition of the band through both experimental and theoretical calculations, we revealed the band hybridization that induced magnetic moment on Sn, through the magnetic exchange interactions in the system. In addition to updating the ARPES spectra and EDC curves of the flat band, we have prepared orbital-resolved bulk band structure calculations to illustrate the hybridization between Mn- d electrons and Sn- p electrons (see Supplementary Materials). This further supports the existence of magnetic exchange interactions between Mn and Sn atoms.

Some minor questions and suggestions:

Comment 8: *What is the meaning of the sentence in Lines 65-67?*

Response: In the original manuscript, lines 65-67 address the contradiction regarding the magnetic ordering in GdMn_6Sn_6 . GdMn_6Sn_6 exhibits ferrimagnetic order, while Mn atoms, confined within the Kagome lattice, possess strong localized magnetism, which is not conducive to forming long-range magnetic order. Although neutron diffraction can detect ferromagnetic coupling between Mn atoms, it is insufficient to explain the mechanism of magnetism formation in GdMn_6Sn_6 . We will revise this section in the updated manuscript to improve the clarity and logical flow.

=>Revised manuscript: *Lines 52-54:* Although neutron diffraction experiments have revealed ferromagnetic coupling within the kagome-Mn layer^{33,34}, the relationship between these electronic states and magnetism remains unclear.

Comment 9: *What is the meaning of "cruising magnetism" in Line 102?*

Response: I apologize for the use of the term "cruising magnetism," which is not standard. To be consistent, it should be changed to "itinerant magnetism," corresponding to "localized magnetism." Itinerant magnetism primarily exists in the $4s$ and $3d$ electrons of transition metals such as iron, cobalt, and nickel, and plays a crucial role in explaining magnetic phenomena in metals.

=>Revised manuscript: This part text of the manuscript has been revised, and the wrong wording has been replaced. ' Itinerant ' is used consistently throughout the manuscript.

Comment 10: *Fig.1(d,e) are too small compared to 1a. Figure 1 should be rearranged.*

Response: Thanks for the referee's suggestion. The scale of Fig. 1(a) is indeed too large, and we will make the necessary adjustments in the revised manuscript.

=>Revised manuscript:

Comment 11: *The 3D view in Fig. 1c is not a good example to show the Dirac point at the K point and vHS near the M point in the band structure. I suggest using the simpler version.*

Response: The 3D band structure diagram appears complex and may not facilitate reader understanding. We will replace it with a 2D schematic in the revised manuscript.

=>Revised manuscript:

Comment 12: *The inset in Fig. 2a is not clear in showing the experimental setup as depicted in Line 128.*

Response: The experimental setup for XMCD and its description in the manuscript are not detailed enough. We will redraw the illustration in Fig.3 and provide a more thorough explanation in the revised manuscript.

=>Revised manuscript:

Comment 13: *Fig. 2d is not clear to show the moment direction. It would be great to show some views along the c-axis to display the orientations of the moments in each layer.*

Response: The magnetic moment diagram in Fig. 2(d) is indeed too crowded, making it difficult for readers to interpret. We will try to add a top view along the *c*-axis to facilitate better understanding.

=>Revised manuscript:

Comment 14: *Line 96, c should be subscripted, like T_c .*

Response: We apologize for not noticing the issue with the subscripts. We will make the necessary corrections in the revised manuscript.

=>**Revised manuscript:** Line 81: “ T_c ”

Comment 15: *Line 127 has a typo: “magnetize”.*

Response: We apologize for the spelling error. We will thoroughly review the revised manuscript to avoid similar issues in the future.

=>**Revised manuscript:** Line 115: “magnetization”.

Comment 16: *Add references for Lines 137-138.*

Response: References are indeed needed here, and we will add the appropriate citations.

=>**Revised manuscript:** (Line 126) Add citations: “*Hossain, M. A., et al., Physical review letters, 101(1), 016404(2008).*”

Comment 17: *Line 141 has a typo: “non-magnetic”.*

Response: Thank you for your meticulous and patient review. We apologize for this spelling error, and we will revise and thoroughly check the manuscript.

=>**Revised manuscript:** Line 130: “non-magnetic”.

Comment 18: *Units are missing in Table I.*

Response: Thank you for your correction. We will add the unit for magnetic moments μB in the revised manuscript.

=>**Revised manuscript:** Table I: Unit: μB .

Summary of changes:

1) **Original manuscript** (page 2, line 28): ‘..... magnetic properties of GdMn_6Sn_6 were thoroughly investigated. Remarkably, we discovered induced magnetic moments.....’

=>**Revised manuscript** (page 2, line 10): We have added a description about the flat band, ‘..... magnetic properties of GdMn_6Sn_6 were thoroughly investigated. We observed highly localized electronic states resulting from spin frustration in the Mn-Kagome lattice. Remarkably, we discovered induced magnetic moments.....’

2) **Original manuscript** (page 2, line 37): ‘Transition metal-based kagome magnets offer a fertile ground for investigating various topological quantum magnetic phases ¹⁻⁷.’

=>**Revised manuscript** (page 2, line 20): We have added a reference in the main text of our paper [D. Guterding, et al, *Scientific reports* 6.1: 25988(2016).]. ‘Transition metal-based kagome magnets offer a fertile ground for investigating various topological quantum magnetic phases ^{1-7,39}.’

3) **Original manuscript** (page 2, line 64): ‘More importantly, the establishment of ferromagnetic order requires the itinerant magnetism of valence band electrons. Neutron diffraction experiments revealing ferromagnetic coupling within the Mn-Mn layer ^{33,34}, yet Mn 3d electrons influenced by the geometric phase of their wave functions, resulting in localization and spin frustration. This localization, as the consequences of the unique kagome configuration, creates a competitive interaction with the itinerant electrons in the formation of magnetically ordered ground states of RMn_6Sn_6 materials.’

=>**Revised manuscript** (page 3, line 47): ‘More importantly, the establishment of long-range magnetic ordering requires the itinerant magnetism of valence band electrons, and the magnetic order is closely related to the unique highly localized flat band generated by the Mn-kagome geometric lattice in RMn_6Sn_6 . This localization competes with the itinerant electrons in the formation of the magnetic ordered ground states in GdMn_6Sn_6 , making the investigation of the electronic states particularly important. Although neutron diffraction experiments have revealed ferromagnetic coupling within the kagome-Mn layer ^{33,34}, the relationship between these electronic states and magnetism remains unclear.’

4) **Original manuscript** (page 2, line 73): ‘However, both experimental and theoretical evidence researching the magnetic interaction mechanisms within the RMn_6Sn_6 family are insufficient, and further exploration is urgently needed, which is critical for investigating their topological properties and the development of applications based on magnetic kagome materials.’

=>**Revised manuscript** (page 3, line 57): We have added a reference in the main text of our paper [Y. Lee, et al, *Physical Review B* 108.4: 045132(2023).]. ‘However, the investigation on the mechanism of magnetic properties, especially the exchange interactions between R-Mn and the magnetic coupling of RMn_6Sn_6 remain are still lacking from either experimental or theoretical aspects ⁴⁰, which is critical for investigating their topological properties and the development of applications based on magnetic kagome materials.’

5) **Original manuscript** (page 5, Fig. 1):

=> **Revised manuscript** (page 4, Fig. 1):

- 6) **Original manuscript** (page 5, the caption of Fig. 1): ‘**Crystal and electronic structures of bulk single-crystal GdMn₆Sn₆.** **a** The crystal structure of GdMn₆Sn₆. Kagome lattice structure constructed by Mn (blue) atoms in the a-b plane, sandwiched between Gd(black) and Sn(red) atoms. **b** Bulk and projected surface BZ of GdMn₆Sn₆ with marked high-symmetry points. **c** The sketch of 3D band structure of kagome lattice from TB model (without SOC). **d** Calculated spin polarized band along the $\bar{M} - \bar{\Gamma} - \bar{M} - \bar{K} - \bar{\Gamma} - \bar{K}$ direction, with majority (blue) and minority(red) spin states. **e** Experimental (left panel) and calculated (right panel) band structure along the $\bar{M} - \bar{K} - \bar{\Gamma} - \bar{K} - \bar{M}$ high symmetry. The flat band appears clearly, which is indicated by a rectangle dash line at about 0.4-0.6 eV below E_F .’

=>**Revised manuscript** (page 4, the caption of Fig. 1): ‘**Crystal and electronic structures of bulk single-crystal GdMn₆Sn₆.** **a** The crystal structure of GdMn₆Sn₆. Kagome lattice structure constructed by Mn (blue) atoms in the a-b plane, sandwiched between Gd(black) and Sn(red) atoms. **b** The sketch of TB model in kagome lattice with NN in-plane hopping. **c** Schematic diagram of 2D band structure of kagome lattice from TB model. **d** Bulk and projected surface BZ of GdMn₆Sn₆ with marked high-symmetry points. **e** Experimental structure along the $\bar{M} - \bar{K} - \bar{\Gamma} - \bar{K} - \bar{M}$ high symmetry. The flat band appears clearly, which is indicated by red arrows. **f** EDC in experiments along $\bar{\Gamma} - \bar{K}$ show the flat band around 0.5 eV below E_F .’

- 7) **Original manuscript** (page 4, line 91): ‘The crystal and electronic structures of GdMn₆Sn₆.} GdMn₆Sn₆ single crystal exhibits a layered hexagonal structure with the space group P6/*mmm* (No. 191), characterized by lattice parameters ($a=b=0.552\text{\AA}$, $c=0.902\text{\AA}$). This structure is composed of manganese kagome planes, with tin (Sn) and gadolinium (Gd) distributed across different layers along the c-axis, as illustrated in Fig. 1a. GdMn₆Sn₆ is defined by its in-plane ferrimagnet ground state (Curie temperature, $T_c = 440$ K), where the Gd-Mn atoms are collinear antiferromagnetically coupled. Fig. 1b displays the bulk and projected surface Brillouin zones (BZ) along with high symmetry points. The unique electronic structure in momentum space can be derived using the tight-binding (TB) model^{15,16}, focusing on nearest neighbor (NN) hopping in the kagome lattice. The results, shown in Fig. 1c, depict the Dirac cone at *K* points, a saddle point at *M* points, and a flat band throughout the entire BZ, indicating strong electron localization and correlation. This contrasts with the cruising magnetism observed in GdMn₆Sn₆, necessitating further analysis of the magnetic formation mechanism in this material.

To probe the topological electronic states associated with the Mn kagome lattice, we performed DFT calculations and ARPES measurements of the band structure in the first BZ at the $k_z = \pi$ plane. Additionally, to take into account strong correlation effects, we used a GGA+U density functional applied for Mn 3*d* states^{27,28}. The value of the effective Hubbard parameter $U^* = U - J = 3$ eV was chosen to reproduce correctly Curie temperature (the same effective U^* was used in a previous study¹⁷). As well, the GGA+U functional was used for treatment of strongly correlated Gd 4*f* states ($U^* = 6$ eV). The calculations were performed using a self-consistent Green function method as it is implemented within the multiple scattering theory²⁹. The resulting spin-polarized electronic structure is depicted in Fig. 1d, with majority and minority spin states represented by blue and red lines, respectively. Our experimental observations of the band structure along the $\bar{M} - \bar{K} - \bar{\Gamma} - \bar{K} - \bar{M}$ direction at a photon energy of $h\nu = 94$ eV in Fig. 1e. There is a clear flat band at 0.4-0.6 eV below the E_F ,

which extending across the entire BZ. This flat band characterized by a high density of states, primarily originates from the Mn $3d - t_{2g}$ orbital electrons localized within the kagome lattice, corroborating with the ARPES results.’

=>Revised manuscript (page 4, line 76): ‘**The crystal and electronic structures of GdMn₆Sn₆.** GdMn₆Sn₆ single crystal exhibits a layered hexagonal structure with the space group P6/*mmm* (No. 191), characterized by lattice parameters ($a=b=5.47\text{\AA}$, $c=8.92\text{\AA}$). This structure is composed of manganese kagome planes, with tin (Sn) and gadolinium (Gd) distributed across different layers along the c -axis, as illustrated in Fig. 1a. GdMn₆Sn₆ is defined by its in-plane ferrimagnet ground state (Curie temperature, $T_c = 440\text{ K}$), where the Gd-Mn atoms are collinear antiferromagnetically coupled. As shown in Fig. 1(b), in a kagome lattice, the wave functions of adjacent sites on a hexagonal ring have opposite phases. This destructive results in a zero transition probability between nearest-neighbor (NN) sites, leading to electron localization within the hexagonal ring. The band structure of the 2D kagome lattice, considering NN hopping and calculated using the tight-binding (TB) model, is depicted in Fig. 1(c). It displays a Dirac cone at the K point, a saddle point at the M point, and a flat band extending across the entire Brillouin Zone (BZ), indicating strong localization and correlation of electrons. To investigate the topological electronic states associated with the manganese kagome lattice, we conducted ARPES measurements and DFT calculations in the $k_z = 0$ plane (see Supplementary Materials). The band structure observed experimentally along the $\bar{M} - \bar{K} - \bar{\Gamma} - \bar{K} - \bar{M}$ direction at a photon energy of $h\nu = 94\text{ eV}$ is shown in Fig. 1(e). Fig. 1(f) presents the Integrated energy distribution curve (EDC) along the $\bar{\Gamma} - \bar{K}$ direction, clearly revealing the flat band located 0.4-0.6 eV below E_F . Orbital-resolved band structure calculations confirm that the flat band originates from Mn $3d$ orbital electrons (see Supplementary Materials) and exhibits orbital hybridization with Sn- $5p$ electrons. In the GdV₆Sn₆ material related to GdMn₆Sn₆, resonant ARPES has also demonstrated the hybridization of V- $3d$ and Sn- $5p$ orbitals, which can promote the formation of long-range magnetic order⁴².’

8) Original manuscript (page 6, Fig. 2):

=>Revised manuscript (page 6, Fig. 2):

9) **Original manuscript** (page 6, the caption of Fig. 2): ‘XAS and XMCD spectra of GdMn₆Sn₆ samples at Gd M_{4,5} edges, Mn L_{2,3} edges and Sn M_{4,5} edges. a-c Experimental XAS and XMCD spectra of Gd, Mn and Sn elements, compared with calculated XAS and XMCD spectra, obtained by reversing of the helicity of x-ray and external magnetic field (± 0.4 T) at 100 K. Labels I⁺ and I⁻ stand for the different polarization directions. A sketch of the experiment is shown in the inset of (a). d Schematic of the microscopic magnetic structure in GdMn₆Sn₆, showing that magnetic moments on Mn are anti-parallel coupled with those on Sn and Gd.’

=>Revised manuscript (page 6, the caption of Fig. 2): XAS and XMCD spectra of GdMn₆Sn₆ samples at Gd M_{4,5} edges, Mn L_{2,3} edges and Sn M_{4,5} edges. a-c Experimental XAS and XMCD spectra of Mn, Gd, and Sn elements, compared with calculated XAS and XMCD spectra. I⁺ and I⁻ represent the absorption intensities of the sample under positive and negative magnetic fields in the XAS spectra, respectively. The XMCD data can be obtained by measuring the difference in the absorption spectra I⁺ and I⁻ of the sample. d A sketch of the experimental configuration diagram. e-f Schematic of the microscopic magnetic structures in GdMn₆Sn₆, showing that magnetic moments on Mn are anti-parallel coupled with those on Sn and Gd.’

10) **Original manuscript** (page 6, the caption of Fig. 2): ‘XAS and XMCD spectra of GdMn₆Sn₆ samples at Gd M_{4,5} edges, Mn L_{2,3} edges and Sn M_{4,5} edges. a-c Experimental XAS and XMCD spectra of Mn, Gd, and Sn elements, compared with calculated XAS and XMCD spectra, obtained by reversing of the helicity of x-ray and external magnetic field (± 0.4 T) at 100 K. Labels I⁺ and I⁻ stand for the different polarization directions. A sketch of the experiment is shown in the inset of (a). d Schematic of the microscopic magnetic structures in GdMn₆Sn₆, showing that magnetic moments on Mn are anti-parallel coupled with those on Sn and Gd.’

=>Revised manuscript (page 6, the caption of Fig. 2): ‘XAS and XMCD spectra of GdMn₆Sn₆

samples at Gd M_{4,5} edges, Mn L_{2,3} edges and Sn M_{4,5} edges. a-c Experimental XAS and XMCD spectra of Mn, Gd, and Sn elements, compared with calculated XAS and XMCD spectra. I⁺ and I⁻ represent the absorption intensities of the sample under positive and negative magnetic fields in the XAS spectra, respectively. The XMCD data can be obtained by measuring the difference in the absorption spectra I⁺ and I⁻ of the sample. **d** A sketch of the experimental configuration diagram. **e-f** Schematic of the microscopic magnetic structures in GdMn₆Sn₆, showing that magnetic moments on Mn are anti-parallel coupled with those on Sn and Gd.'

11) **Original manuscript** (page 5, line 125): 'Fig. 2a inset schematically presents the experimental setup, employing circularly polarized light of positive (blue) and negative (red) polarization. Since the easy magnetize direction is in-plane, we aligned the incident x-ray beam at a 60-degrees angle relative to the sample surface normal. The experiments were conducted at 100 K.'

=>**Revised manuscript** (page 5, line 104): 'XMCD experiments are based on the principle of magnetic circular dichroism. Magnetic samples exhibit different absorption intensities under different polarized light due to optical selection rules. This phenomenon arises because polarized light breaks the system symmetry, leading to selective excitation of electronic states. Experimentally, switching the direction of the magnetic field and the direction of polarization is equivalent. The magnetic field changes the electronic states through Zeeman splitting, thereby producing a difference in the absorption spectra, which is used to analyze the magnetic properties of the material. Fig. 2(d) illustrates the experimental setup. The XAS spectra of the sample are obtained using circularly polarized light with positive (black) and negative (red) polarization or by applying an external magnetic field parallel (or antiparallel) to the direction of the light beam. The difference between these spectra constitutes the XMCD spectrum. Since the easy magnetization direction is in-plane, we aligned the incident x-ray beam at a 60-degrees angle relative to the sample surface normal direction. The experiments were conducted at 100 K.'

12) **Original manuscript** (page 6, line 137): 'These peak structures and positions are consistent with those in other Mn-based materials.'

=>**Revised manuscript** (page 7, line 126): We have added a reference in the main text of our paper [A. M. Hossain, et al. *Physical review letters* 101.1: 016404(2008).]. 'These peak structures and positions are consistent with those in other Mn-based materials ⁴¹.'

13) **Original manuscript** (page 7, line 156): 'The Sn1-site is closest to the Mn kagome layer and thus has the largest contribution to the induced magnetic moments, which suggests an exchange interaction between the Mn 3*d* and Sn 5*p* electrons.'

Finally, we draw a schematic diagram of the microscopic magnetic structure of GdMn₆Sn₆ based on the experimentally magnetic coupling directions of different elements as shown in Fig. 2d, in which the magnetic moments of Gd *f* and Sn *p* electrons are antiparallel coupled with Mn *d* electrons. Such complex magnetic structures suggest that the hybridization of valence band electrons may play a crucial role in elucidating the origin of the magnetic mechanism of GdMn₆Sn₆.'

=>**Revised manuscript** (page 7, line 145): 'We have illustrated the microscopic magnetic

structure of GdMn_6Sn_6 based on the experimental magnetic coupling directions of different elements, as shown in Figs. 2(d) and 2(e). The magnetic moments of Gd f electrons and Sn p electrons couple antiferromagnetically with Mn d electrons. In a unit cell, the Sn1, Sn2, and Sn3 atoms are each in different chemical environments. The Sn1 site is sandwiched between the upper and lower Mn kagome layers, allowing it to directly participate in the Mn-Sn-Mn magnetic exchange interaction⁴⁴. It is also spatially closer to the Mn atoms, resulting in the largest induced magnetic moment. Although the Sn2 site is close to the Mn kagome layer in the c direction, it is pushed to the outside of the unit cell by the Gd atoms, thus reducing its induced magnetic moment. The Sn3 atoms are influenced by the Gd atoms, which weakens their induced magnetic moments. This complex magnetic structure indicates that the hybridization of valence band electrons may play a crucial role in elucidating the origin of the magnetic mechanism in GdMn_6Sn_6 .

14) **Original manuscript** (page 7, the title of table I): ‘Magnetic Moments of Mn and Sn Atoms at Various Sites from DFT Calculations.’

=>**Revised manuscript** (page 7, the title of table I): ‘Magnetic Moments (μ_B) of Mn and Sn Atoms at Various Sites from DFT Calculations.’

15) **Original manuscript** (page 8, line 167): ‘For calculations we utilized the same DFT functional as for calculations of electronic and magnetic structure mentioned above.’

=>**Revised manuscript** (page 8, line 159): ‘Understanding the strength of interactions within the kagome lattice is essential for unraveling the magnetic properties of this material. Due to the weak SOC effects in GdMn_6Sn_6 and the isotropic exchange interactions within the Mn layers, we adopt the Heisenberg model to describe the spin correlations instead of Kitaev model⁴³.’

16) **Original manuscript** (page 9, Fig. 3):

=>**Revised manuscript** (page 9, Fig. 3): We adjusted the font size and contrast of Fig. 3.

17) **Original manuscript** (page 9, the caption of Fig. 3): ‘The distribution map of magnetic exchange constants J (in meV) in GdMn_6Sn_6 . **a-c** Without considering the Coulomb interaction U^* . **d-f** With considering the Coulomb interaction U^* . The exchange coupling strength in intralayer kagome Mn-Mn, interlayer Mn-Mn and Gd-Mn within a supercell were calculated respectively. The positive and negative values of J are represented by red and blue regin.’

=>**Revised manuscript** (page 9, the caption of Fig. 3): ‘The distribution map of magnetic exchange constants J (in meV) in GdMn_6Sn_6 . **a-c** Without considering the Coulomb interaction U . **d-f** With considering the Coulomb interaction U . The exchange coupling strength in intralayer kagome Mn-Mn, interlayer Mn-Mn and Gd-Mn within a supercell were calculated respectively. The positive (ferromagnetic) and negative (antiferromagnetic) values of J are represented by red and blue regin.’

18) **Original manuscript** (page 8, line 178): ‘Fig. 3 compare the exchange interaction images without/with consideration of the Hubbard U correction, respectively.’

=>**Revised manuscript** (page 8, line 171): ‘Fig. 3 shows the exchange interaction strength J (in meV) between intralayer and interlayer Mn-Mn, Gd-Mn atoms in the real-space supercell, without and with the Hubbard U correction, respectively, where the axes (in unit of \AA) indicate the relative lattice position in real space from the central atom. The color scale in red and blue represent positive (ferromagnetic coupling) and negative (antiferromagnetic coupling) values of J between this site atom with the central atom, respectively.’

19) **Original manuscript** (page 8, line 186): ‘It is noteworthy that the Coulomb interaction, U , induces a rotation of the C_2 symmetry axis of J , which is intimately linked to the correlated

interactions among electrons. Charge transfer between Mn atoms leads to the hopping of electrons with the same spin between different energy levels. The exchange integral intensity J changes from negative to positive at this time, resulting in a ferromagnetic coupling between Mn atoms within the kagome layer, and reducing the absolute value of the exchange interaction intensity.'

=>Revised manuscript (page 9, line 184): 'Notably, the Coulomb interaction U causes the exchange integral strength between Mn atoms to shift from negative to positive due to the charge transfer between Mn atoms, while the distribution of J maintains C_2 symmetry. Electrons with the same spin transition between different energy levels among the Mn atoms, leading to ferromagnetic coupling between Mn atoms within the kagome layer and reducing the absolute value of the exchange interaction strength.'

20) Original manuscript (page 10, Fig. 4):

=>Revised manuscript (page 11, Fig. 4):

21) **Original manuscript** (page 10, the caption of Fig. 4): ‘**Schematic illustrations of exchange interactions in GdMn₆Sn₆.** **a** Spin-polarized Fermi surface calculation. **b** Mechanism of magnetic interaction. **c** *p-d* orbital hybridization between Mn-Sn atoms. **d** Magnetic moments and exchange interactions between Gd-Mn atoms.’

=>**Revised manuscript** (page 11, the caption of Fig. 4): ‘**Schematic illustrations of exchange interactions in GdMn₆Sn₆.** **a** A sketch of mechanism of magnetic interaction from calculated density of states (DOS). **b** *p-d* orbital hybridization between Mn-Sn atoms. **c** Magnetic moments and exchange interactions between Gd-Mn atoms.’

22) **Original manuscript** (page 10, line 209): ‘Further, we calculated the spin-polarized Fermi surface, as shown in Fig. 4a. It demonstrates C_2 symmetry, corresponding to the symmetry observed in the exchange interaction intensity diagram in Fig. 3. We suggested this symmetry break is a result of magnetism.’

=>**Revised manuscript:** As the original manuscript's discussion on spin-polarized Fermi surfaces contributed minimally to the main point of the article, it has been removed.

23) **Original manuscript** (page 10, line 216): ‘Such hybridization, through its covalent mechanism, diminishes local magnetization and system energy, thereby leading to the antiparallel alignment observed in our XMCD experiments.’

=>**Revised manuscript** (page 10, line 210): ‘Such hybridization, through its covalent mechanism, diminishes local magnetization and system energy, thereby leading to the antiparallel alignment observed in our XMCD experiments. Simultaneously, using the Disordered Local Moment (DLM) method to simulate the magnetic properties of GdMn₆Sn₆ in a high-temperature disordered state also verified the presence of indirect double exchange interaction. In the DLM model, thermal perturbations are introduced, assuming that the magnetic moments in the system are disordered, meaning that the direction of each atom's magnetic moment is random. This allows us to obtain the electronic structure and energy distribution in the disordered magnetic state. By comparing the total energy and atomic magnetic moment distribution at T=0 K with those calculated using the DLM method, it was found that the system tends to form a magnetically ordered state at low temperatures, which is consistent with the ferromagnetic ordering induced by double exchange interaction.’

24) **Original manuscript** (page 12, line 280): ‘XMCD reveals the differences in the absorption rates of left- and right-circularly polarized x-ray photons. The XMCD experiments possess the element-selective capabilities, as evidenced by the photon energy at the absorption edges. In these experiments, the effect of left and right circularly polarized light was emulated by switching the direction of the magnetic field. This work was conducted at the soft x-ray beamline BL07U of SSRF. All samples were cleaved *in situ* under a high (better than 10⁻⁵ Pa) vacuum and subsequently transferred to the measurement chamber (better than 10⁻⁶ Pa) equipped with eight vector magnets.’

=>**Revised manuscript** (page 13, line 286): ‘XMCD reveals the differences in the absorption rates of left- and right-circularly polarized x-ray photons due to selection rules. Since the energy depends on specific elements, XMCD can achieve element selectivity by choosing an appropriate

photon energy range. In these experiments, the effect of left and right circularly polarized light was emulated by switching the direction of the magnetic field. The intensity of the photocurrent is measured to characterize the absorption rate of x-ray under positive and negative magnetic field, which namely the TEY method. Due to the shallow penetration depth of soft x-ray photons, the probing depth of the TEY method is limited to a few tens of nanometers below the surface. This work was conducted at the soft x-ray beamline BL07U of SSRF. All samples were cleaved *in situ* under a high (better than 10^{-5} Pa) vacuum and subsequently transferred to the measurement chamber (better than 10^{-6} Pa) equipped with eight vector magnets, capable of reaching a maximum magnetic field of 0.8 T. To eliminate systematic machine errors, all data were measured multiple times and averaged.'

25) **Original manuscript** (page 13, line 288): 'Electronic band structure calculations were performed within the framework of DFT using the Vienna *ab initio* simulation package (VASP)³⁵, employing the projector augmented wave (PAW) method. The generalized gradient approximation (GGA) of the Perdew-Burke-Ernzerhof (PBE) method was used for the exchange-correlation functionals³⁶. Additionally, SOC effect was incorporated in the calculations. the spin-orbit coupling (SOC) effect was incorporated in the calculations. The kinetic energy cutoff for the plane-wave basis was set to at 380 eV, and a $12 \times 12 \times 6$ *k*-mesh was selected with an accuracy of 10^{-8} eV in the calculations. The electronic correlation effects among Mn 3*d* electrons were addressed using the GGA plus Hubbard U method³⁷. The Bulk states were calculated by constructing localized Wannier functions with the WANNIERTOOLS package³⁸.'

=>**Revised manuscript** (page 13, line 299): 'Electronic band structure calculations were performed within the framework of DFT using the Vienna *ab initio* simulation package (VASP)³⁵, employing the projector augmented wave (PAW) method. The calculations were also performed using a self-consistent Green function method as it is implemented within the multiple scattering theory²⁹. Spin-orbit coupling (SOC) effects have been included in the calculations. The generalized gradient approximation (GGA) of the Perdew-Burke-Ernzerhof (PBE) method was used for the exchange-correlation functionals³⁶. Additionally, SOC effect was incorporated in the calculations. the spin-orbit coupling (SOC) effect was incorporated in the calculations. The kinetic energy cutoff for the plane-wave basis was set to at 380 eV, and a $12 \times 12 \times 6$ *k*-mesh was selected with an accuracy of 10^{-8} eV in the calculations. To take into account strong correlation effects, we used a GGA+U density functional applied for Mn 3*d* states^{27, 28, 37}. The value of the effective Hubbard parameter $U_{Mn}^* = U - J = 4 - 1 = 3$ eV was chosen to reproduce correctly Curie temperature (the same effective U^* was used in a previous study^{17, 40}). As well, the GGA+U functional was used for treatment of strongly correlated Gd 4*f* states ($U_{Gd}^* = 6$ eV).'

RESPONSES TO REVIEWERS' COMMENTS:

To Reviewer #1:

Comment: *The authors have made their explanation to all reviewer' comments, mostly convincing, also highlighted the significance of their work by serving half-filled 4f band Gd as an ideal model of studying the magnetic exchange interaction in RMn₆Sn₆. The authors also improved their data quality (APRES, etc.), clarified the importance of flat band signature in magnetic coupling magnetism, and provided a more detailed discussion of different sites Sn. The statement in the revised manuscript is more clear to me. Therefore, I would recommend this manuscript to publish in Communications Physics.*

Response: We sincerely thank the reviewer for the positive and thoughtful comments. We have revised the manuscript according to your suggestions, which greatly helped us improve its quality. We are honored by your recommendation for publication in Communications Physics and thank you again for your time and effort in reviewing our work.

To Reviewer #2:

Comment: *The authors addressed the questions and comments from the referees well. However, one concern remains: the ARPES data does not support their main claim. Why not highlight the hybridization of the V case at the end of the paper and suggest possible connections and future studies?*

Response: We sincerely thank the reviewer for the valuable and encouraging comments on improving our manuscript. We agree that the current ARPES results do not provide the most direct evidence of orbital hybridization. However, our DFT calculations strongly support the hybridization between Mn *d* and Sn *p* electrons, which induces magnetic moments on Sn atoms. As shown in the Supplementary Fig.4, the orbital-resolved band structure calculations clearly reveal significant overlap between Mn *d* and Sn *p* orbitals, indicating robust hybridization. We also agree with your suggestion that future resonant ARPES experiments could directly verify this hybridization, so we will revise the conclusions to emphasize the importance of the hybridization effect and propose resonant ARPES as a promising direction for future studies in RMn₆Sn₆. We look forward to your feedback on the revised version.

=>Revised manuscript: Line 279-296: To conclude, we conducted a comprehensive investigation of GdMn₆Sn₆ magnetic and electronic structure by using photoemission spectroscopy and DFT calculations. Our study demonstrated that localized kagome flat band is primarily attributed to the Mn-*e_g* orbitals. Our experiments revealed the ferrimagnetic structure of GdMn₆Sn₆ by element-resolved XMCD and detected the magnetic moments in the nonmagnetic element Sn. Theoretical calculations highlighted double exchange interaction between Sn 5*p* and Mn 3*d* electrons, leading to the ferromagnetic coupling within Mn-Mn atoms, and the interlayer Gd-Mn atoms antiferromagnetic coupling resulting from the indirect interaction between rare-earth *f* electrons and Mn *d* electrons. Moreover, the orbital-resolved band structure calculations confirm the hybridization between Mn 3*d* and Sn 5*p* orbitals that plays a critical role in the magnetic coupling of GdMn₆Sn₆, which has been demonstrated in similar material, GdV₆Sn₆, by

resonant ARPES. Future experiments could provide more direct evidence of orbital hybridization in the RMn_6Sn_6 family. These results provide possibilities to explore the origin of the microscopic magnetic mechanism in the kagome magnets and offer potential connections to magnetic and topological properties in QAHE complex quantum systems. Furthermore, our research provides theoretical support and experimental evidence for advancements in applications such as spintronic devices and quantum computing, with the potential to drive the development of these technologies.

To Reviewer #3:

Comment: *I appreciate the authors' detailed explanation, especially regarding the XMCD and XAS data. The ARPES data appears much clearer than in the previous version. The revised manuscript has been significantly improved in terms of clarity. The magnetic moment on the Sn site induced by double exchange interaction could be of potential interest to the community.*

However, I noticed that the references are incorrectly sorted in the revised version. For example, in Line 21, the references are listed as 1-7, followed by 39. Additionally, in Line 53, Refs. 33 and 34 are theory papers, but the authors cite them as experimental evidence in response to Referee 1's question. The authors should carefully check the manuscript. I can recommend publication after these minor corrections.

Response: We sincerely thank the reviewer for the thorough review and positive feedback on our responses and manuscript. We are pleased that the additional details on the XMCD and XAS experiments, as well as the improvements in the visualization of the ARPES data, have been recognized.

Regarding the reference issues you pointed out, we have carefully revised the manuscript, using BibTeX to automatically organize the reference order and correct the previous errors, ensuring all citations are now accurate. We sincerely apologize for the earlier mistakes. Furthermore, we have conducted a more comprehensive and detailed review of the manuscript, addressing other issues as well. We hope that the revised manuscript will receive your recommendation for publication.